# Cryo-EM structures of apo-APC/C and APC/C^CDH1:EMI1 complexes provide insights into APC/C regulation

Anna Höfler [1,5], Jun Yu [1,5], Jing Yang[2], Ziguo Zhang [2], Leifu Chang [2], Stephen H. McLaughlin [2], Geoffrey W. Grime [3], Elspeth F. Garman [4], Andreas Boland [1] ✉ & David Barford [2] ✉

APC/C is a multi-subunit complex that functions as a master regulator of cell division. It controls progression through the cell cycle by timely marking mitotic cyclins and other cell cycle regulatory proteins for degradation. The APC/C itself is regulated by the sequential action of its coactivator subunits CDC20 and CDH1, post-translational modifications, and its inhibitory binding partners EMI1 and the mitotic checkpoint complex. In this study, we took advantage of developments in cryo-electron microscopy to determine the structures of human APC/C^CDH1:EMI1 and apo-APC/C at 2.9 Å and 3.2 Å resolution, respectively, providing insights into the regulation of APC/C activity. The high-resolution maps allow the unambiguous assignment of an α-helix to the N-terminus of CDH1 (CDH1^α1) in the APC/C^CDH1:EMI1 ternary complex. We also identify a zinc-binding module in APC2 that confers structural stability to the complex, and we confirm the presence of zinc ions experimentally. Finally, due to the higher resolution and well defined density of these maps, we are able to build, aided by AlphaFold predictions, several intrinsically disordered regions in different APC/C subunits that likely play a role in proper APC/C assembly and regulation of its activity.

The anaphase-promoting complex/cyclosome (APC/C) is a large multi-subunit E3 ubiqutin ligase that regulates transitions through the cell cycle by controlling the defined degradation of specific cell cycle regulators through the ubiquitin-proteasome system (UPS). APC/C activity and substrate selection are controlled at various levels to ensure the correct order of specific cell cycle events. These regulatory mechanisms include the binding of cell cycle specific coactivator subunits (CDC20 and CDH1), reversible APC/C and coactivator phosphorylation, inhibitory complexes and proteins (for example the mitotic checkpoint complex (MCC) and early mitotic inhibitor 1 (EMI1)), and APC/C SUMOylation. APC/C function, mechanisms and structures have been reviewed extensively[1–4].

In mitosis the APC/C is activated by phosphorylation and by the binding of a coactivator subunit CDC20, to form the APC/C^CDC20 complex that functions to ubiquitinate securin and cyclin B. APC/C-mediated destruction of both securin and cyclin B activates separase to cleave the cohesin ring, thereby triggering sister chromatid segregation and the onset of anaphase. During this transition, the APC/C and CDH1 are dephosphorylated, resulting in a switch from APC/C^CDC20 to APC/C^CDH1. In addition to mediating the metaphase to anaphase transition, the CDH1-activated APC/C regulates the events of cytokinesis and the entry into G1.

The APC/C binds its substrates through destruction motifs or degrons: the D box, the ABBA motif and the KEN box[5–8]. These motifs

[1]Department of Molecular and Cellular Biology, University of Geneva, Geneva, Switzerland. [2]MRC Laboratory of Molecular Biology, Cambridge CB2 0QH, UK. [3]Ion Beam Centre, Advanced Technology Institute, University of Surrey, Guildford, Surrey GU2 7XH, UK. [4]Department of Biochemistry, University of Oxford, Dorothy Crowfoot Hodgkin Building, South Parks Road, Oxford OX1 3QU, UK. [5]These authors contributed equally: Anna Höfler, Jun Yu. ✉e-mail: Andreas.Boland@unige.ch; dbarford@mrc-lmb.cam.ac.uk

interact with their cognate receptors on two structurally related WD40-domain coactivator subunits (CDC20 in mitosis and CDH1 in G1) that function to recruit substrates to the APC/C. Control of different cell cycle phases by the APC/C is mediated through the different substrate specificities of APC/C[CDC20] and APC/C[CDH1]. Substrate specificity of APC/C[CDC20] is augmented by the MCC which inhibits APC/C[CDC20] activity towards cyclin B and securin[9,10], but not cyclin A[11,12]. Two conserved motifs in both coactivators, a seven residue-N-terminal C box[13,14], and a C-terminal Ile-Arg motif (IR tail)[14,15], are necessary for coactivators to bind the APC/C. The APC/C is a cullin-RING E3 ubiquitin ligase, with the catalytic module composed of the cullin-domain APC2 subunit and the RING-domain APC11 subunit. In humans, two E2 enzymes (UBCH10/UBE2C and UBE2S) bind the catalytic module of the APC/C to catalyse subunit polyubiquitination[16–21]. As well as recruiting substrates, coactivators directly stimulate the catalytic activity of vertebrate APC/C[22] by inducing a conformational change of the catalytic module that permits UBCH10/UBE2C binding[23]. In vertebrates, EMI1 functions as an antagonist of APC/C[CDH1] during S phase and G2 to allow cells to commit to DNA replication and cell division[24–26].

We had previously determined cryo-EM structures of human APC/C[CDH1:EMII] (ref. 19) and unphosphorylated apo-APC/C[27], both at 3.6 Å resolution. Here, by taking advantage of the latest improvements in cryo-EM hardware and software, we determined the structures of APC/C[CDH1:EMII] and apo-APC/C at 2.9 Å and 3.2 Å resolution, respectively. The well-defined cryo-EM maps, as well as currently existing APC/C models combined with AlphaFold predictions[28,29], enabled us to build the most complete and accurate structural models of the APC/C available to date. These models reveal insights into APC/C regulation during mitosis, providing a more detailed structural rational for the phosphorylation-dependent switch from APC/C[CDC20] to APC/C[CDH1], the identification of a zinc-binding module in APC2, and shed light on the role of intrinsically disordered regions (IDRs) in APC/C complex assembly.

## Results

### Structure determination

Our earlier cryo-EM reconstruction of the APC/C in complex with the coactivator CDH1 and the inhibitor EMI1 (APC/C[CDH1:EMII]) (PDB 4UI9) at 3.6 Å resolution[19], and a 3.6 Å resolution structure of the apo unphosphorylated APC/C[27] were determined using data collected on a Polara electron microscope equipped with a Falcon 2 detector. For the current study, we collected data on an APC/C[CDH1:EMII] sample using a Titan Krios microscope with a Falcon 3 detector. These data, together with the use of recently developed refinement methods, detailed in Methods, resulted in reconstructions of the ternary APC/C[CDH1:EMII] complex at 2.9 Å resolution (Fig. 1), providing better definition of the numerous

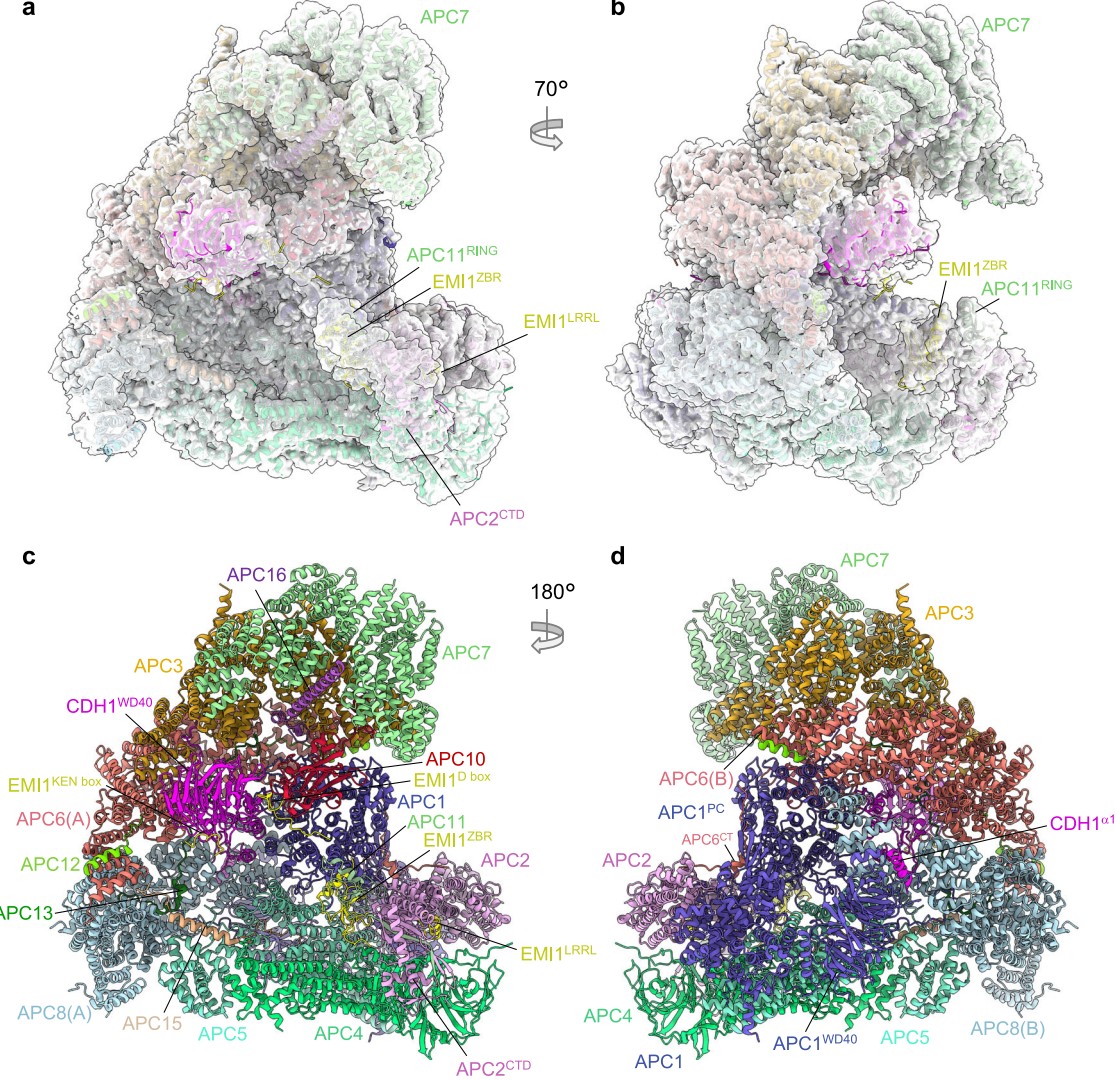

**Fig. 1 | Overall structure of the APC/C[CDH1:EMII] complex. a, b** Two views of the APC/C[CDH1:EMII] ternary complex fitted into the 2.9 Å cryo-EM map. **c, d** Two views of the APC/C[CDH1:EMII] ternary complex shown as ribbon representations.

extended and loop structures that promote inter-subunit interactions, the discovery of a zinc-binding module in APC2, and definition of how the N-terminal α-helix of CDH1 binds to the APC1 subunit.

## Overall features of APC/C[CDH1:EMI1]

For this study, the APC/C[CDH1:EMI1] ternary complex and cryo-EM grids were prepared as described previously[19] (Supplementary Fig. 1). The cryo-EM map refinement work-flow, based on a multi-body approach, is detailed in Methods (Supplementary Figs. 1 and 2 and Supplementary Table 1). Three-dimensional classification generated eight 3D classes, two belonging to the APC/C[CDH1:EMI1] ternary complex and two to the apo-APC/C state (that is without CDH1 and EMI1). The four remaining classes (classes 3-6; Supplementary Fig. 2) corresponded to incomplete APC/C assemblies. The APC/C[CDH1:EMI1] ternary complex map was divided into three masks, with some overlap, for particle subtraction and focussed 3D refinement. This process greatly improved cryo-EM densities in the more peripheral and flexible regions of the map. Specifically, well defined cryo-EM density is visible for the flexible catalytic module comprising the APC2 C-terminal domain (APC2[CTD]), APC11 and the zinc-binding region (ZBR) of EMI1 (EMI1[ZBR]) (Fig. 1a, b and Supplementary Fig. 1c). Additionally, the APC7 homodimer, capping the TPR lobe, is well defined. The current apo-APC/C structure at 3.2 Å supersedes the phosphorylated and unphosphorylated apo-APC/C maps at 3.4 Å and 3.8 Å, respectively[27].

As outlined in Supplementary Table 2, these higher resolution maps allowed marked improvements to regions of the structure compared to the previous 3.6 Å map[19] (PDB 4UI9) according to three main criteria – (i) rebuilding of poorly defined loops; (ii) the fitting of side chains and hence residue assignment to previously built poly-Ala regions; (iii) de novo building of specific regions of the molecule. Four rebuilt or de novo built regions have biological implications and are discussed in more detail below.

## An N-terminal α-helix of metazoan CDH1 interacts with APC1

In the PDB code 4UI9 structure[19], we built a poly-Ala α-helix in density adjacent to the WD40 domain of APC1 (APC1[WD40]), assigned to residues 282-295 of APC1. In the maps reported herein, the clearly defined side chain densities do not fit the APC1 sequence, but instead match the extreme N-terminal sequence of CDH1 (residues 1-19) (Fig. 2a–c). Secondary structure predictions, supported by an AlphaFold model[28,29], strongly confirm this region of human CDH1 to form an α-helix. This places the N-terminus of CDH1 on the outer surface of the APC/C. The CDH1 polypeptide chain then threads through a fenestration at the interface of APC6 and APC8 before forming the C-box motif that engages the C-box binding site of APC8 (Fig. 2a). The remainder of CDH1, including the substrate-recognition degron-binding sites of CDH1[WD40], is positioned within the central cavity of the APC/C (Fig. 1c). Multiple sequence alignments (MSA) indicate CDH1[α1] is highly conserved in metazoan CDH1 (Fig. 2f), supported by its prediction in AlphaFold models of metazoan CDH1[29]. In contrast, the sequence corresponding to human CDH1[α1] is neither conserved in budding yeast CDH1 nor is an N-terminal α-helix predicted in AlphaFold models of budding yeast CDH1[29]. This is consistent with the 3.5 Å structure of the S. cerevisiae APC/C[CDH1:Hsl1] complex[30] that lacks cryo-EM density equivalent to CDH1[α1] of human APC/C[CDH1:EMI1]. The amphipathic CDH1[α1] helix interacts through non-polar interactions with a hydrophobic groove in the APC1[WD40] domain via its N-terminal Met1, as well as Tyr5, Leu9 and Ile13 residues on one side of CDH1[α1] (Fig. 2b, c). Residues Asp2, Asp4, Tyr5, Arg8, Arg11 and Gln12 on the opposite side of CDH1[α1] form specific side chain-side chain interactions with the APC1[WD40] domain as well as with APC8(B). Here, Arg8 plays a central role to anchor CDH1[α1] by deeply inserting into an interface formed by both APC/C subunits APC1 and APC8(B).

AlphaFold models indicated that metazoan CDC20 also possesses a similar N-terminal α-helix to CDH1. Supporting this proposal, a reinterpretation of the 3.2 Å resolution cryo-EM maps of the phosphorylated APC/C[CDC20:CyclinA2] structure[12] (PDB: 6Q6H, EMD-4466), reveals cryo-EM density corresponding to the CDC20 N-terminal α-helix (Fig. 2d). Compared to CDH1[α1] of the APC/C[CDH1:EMI1] ternary complex, CDC20[α1] is shorter, and density corresponding to CDC20[α1] is less well defined, possibly indicating lower occupancy, although we note the lower resolution of the APC/C[CDC20:CyclinA2] map may contribute to this less well defined cryo-EM density.

The position of CDH1[α1] abuts the anchor points of the long 300s loop of APC1 (APC1[300s]) that is inserted within APC1[WD40] (Fig. 2b). The tip of APC1[300s] comprises the APC1 auto-inhibitory segment (APC1[AI]) that in the unphosphorylated APC/C binds the C-box recognition site of APC8, thereby inhibiting CDC20, but not CDH1, binding to unphosphorylated APC/C[27,31]. As reported previously[27,31,32], phosphorylation of APC1[300s] removes APC1[AI] from the C-box recognition site, allowing CDC20 to bind. It seems likely that phosphorylation of APC1, whilst releasing APC1[AI] from the C-box binding site, would also displace the APC1[300s] anchor points, facilitating α1 helix binding. Indeed, comparing the 3.2 Å resolution unphosphorylated apo-APC/C structure determined in this study (discussed below) with APC/C[CDH1:EMI1] shows that CDH1[α1] overlaps with the C-terminal region of APC1[300s] (residues 400-402) as observed in the apo state of APC/C. When bound to CDH1, this loop segment is pushed upwards towards APC8 (Fig. 2c).

CDH1[α1] is a few residues longer than CDC20[α1] (Fig. 2b, d, f, g) offering additional contact sites with APC1. Thus, phosphorylation of APC1[300s] by CDK1/2 remodels APC1[300s] both to remove APC1[AI] from the C-box binding site in APC8(B), and to remove a steric obstacle to the binding of the coactivator α1 helix to APC1[WD40]. We speculated that the larger interface formed between CDH1[α1] and APC1[WD40] compared with CDC20[α1] might contribute to the enhanced affinity of CDH1 for unphosphorylated APC/C relative to CDC20. To assess the relative roles of CDH1[α1] and CDC20[α1] in contributing to the affinity of their respective coactivator to both unphosphorylated and phosphorylated APC/C, we generated a chimeric CDH1 mutant substituting CDC20[α1] for CDH1[α1] (CDH1[CDC20α1]) and assayed APC/C E3 ligase activity (Supplementary Fig. 3a, b). A chimeric mutant of CDC20 (CDC20[CDH1α1]: CDH1[α1] substituted for CDC20[α1]) was unstable. The CDH1[CDC20α1] coactivator however, stimulated the same ubiquitination activity for both unphosphorylated and phosphorylated APC/C, similar to CDH1. This indicated that the α1 helix is not a primary factor in determining the higher activity of CDH1 for unphosphorylated APC/C relative to CDC20. We showed previously that differences between CDH1 and CDC20 in both their N-terminal domains and IR tails contribute to the higher affinity of CDH1 for unphosphorylated APC/C[27].

## Metazoan APC2 contains a zinc-binding module (APC2[ZBM])

An important insight from the higher resolution maps is the discovery of a zinc-binding module in cullin repeat 2 (CR2) of APC2 (APC2[ZBM]). The cryo-EM density, aided by an AlphaFold model of human APC2, allowed a partially built region of APC2 in PDB 4UI9 to be modelled (residues 187-233, Supplementary Table 2) (Fig. 3a–c). This segment constitutes a 45-residue insert between the A and B α-helices of CR2. The insert comprises a 15-residue loop and an exposed α-helix (helix-AB, Fig. 3b), followed by a β-hairpin loop that abuts the N-terminus of the adjoining α-helix B. Four cysteine residues, two from the β-hairpin (Cys221 and Cys224), and two from the distorted N-terminus of α-helix B (Cys231 and Cys233), coordinate a metal ion organized with tetrahedral geometry. The sulphur-metal ion distances measure 2.3 Å on average, typical of sulphur-metal coordination bond lengths. To identify the metal composition, we subjected the APC2:APC11 heterodimer to proton-induced X-ray emission spectroscopy (PIXE)[33]. This revealed APC2:APC11 to contain zinc ions, with no other metal ions. Human APC11 contains three zinc-binding sites. Calibration of the Zn:S stoichiometry indicated 2.5 zinc ions bound to APC2:APC11, less than the four expected assuming a zinc-binding site in APC2 (Table 1).

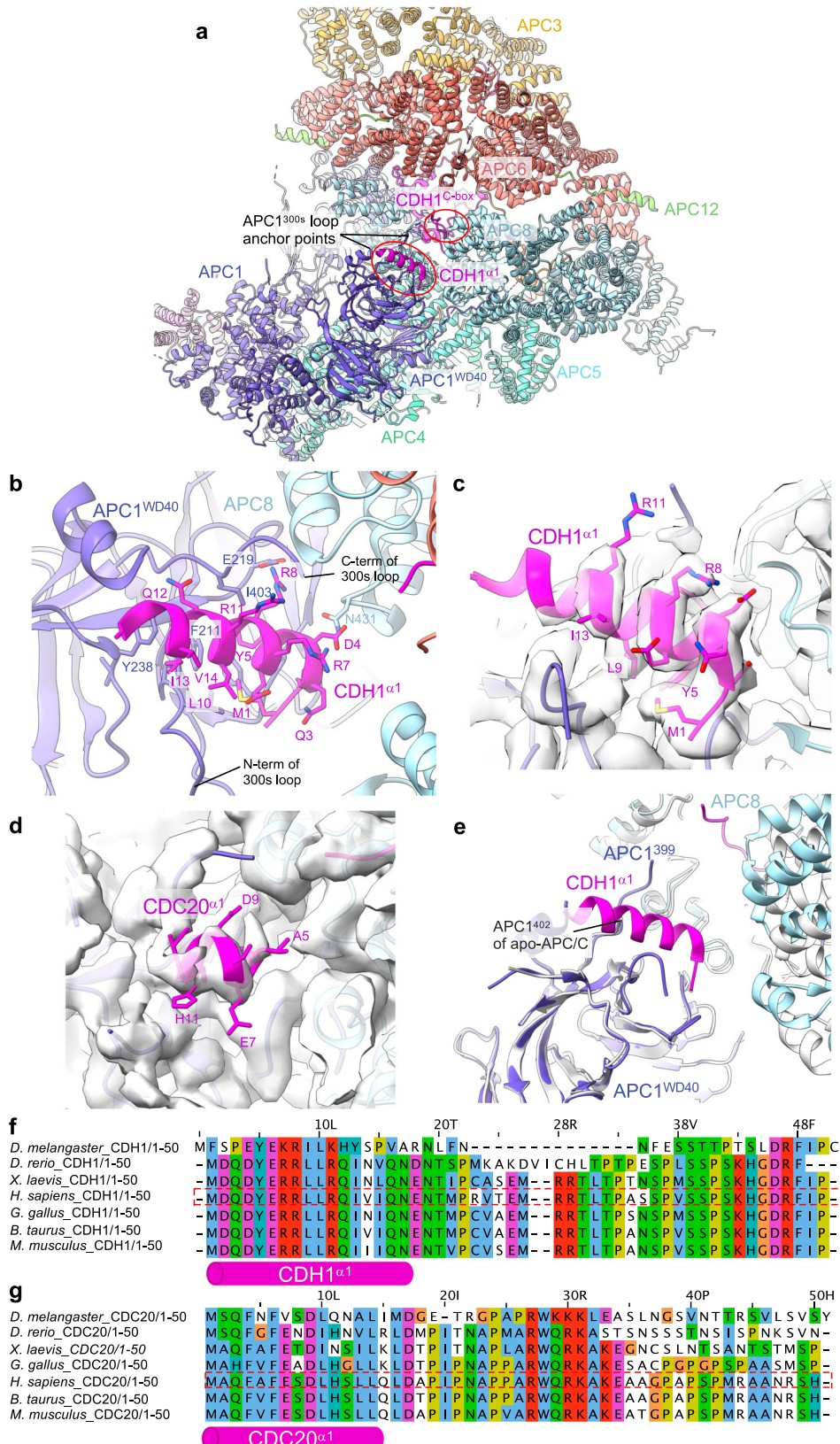

**Fig. 2 | Details of CDH1$^{α1}$ interaction with APC1$^{WD40}$. a** Overview of CDH1$^{α1}$:APC1$^{WD40}$ interactions. **b** Detailed view of CDH1$^{α1}$ interacting with APC1$^{WD40}$. **c** Cryo-EM density map showing CDH1$^{α1}$. **d** An equivalent, although shorter N-terminal α-helix exists in CDC20. Map of CDC20$^{α1}$:APC1$^{WD40}$ interactions (from EMDB: 4466). CDC20$^{α1}$ is based on the AlphaFold model of human CDC20[29].

**e** Superimposition of APC/C$^{CDH1:EMI1}$ onto apo-APC/C showing how CDH1$^{α1}$ overlaps with the C-terminal region of the APC1$^{300s}$ loop (labelled as APC1$^{402}$) of apo-APC/C (coloured light grey). Multiple sequence alignments showing conservation of metazoan CDH1$^{α1}$ (**f**) and CDC20$^{α1}$ (**g**).

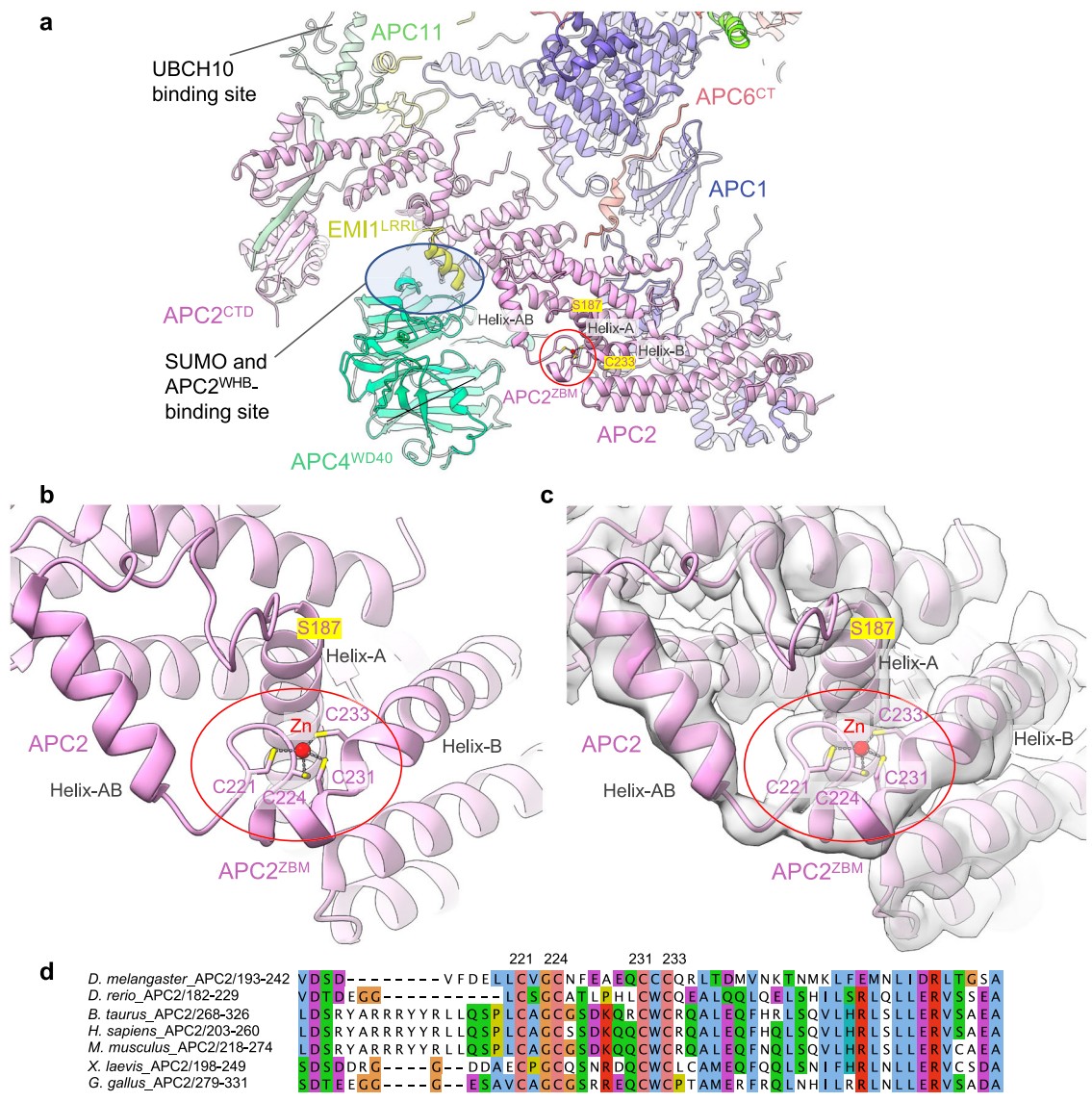

**Fig. 3 | Zn-binding motif of APC2 (APC2^ZBM).** **a** Overview showing APC2^ZBM. **b** Details of the APC2^ZBM (PDB: 9GAW). **c** Cryo-EM density map of APC2^ZBM (EMD-51190). **d** MSA of metazoan APC2 showing conservation of the APC2^ZBM.

However, mutation of the four Cys residues (C221, C224, C231, C233) of the metal-binding site of APC2 to alanines decreased the zinc content by one zinc atom, consistent with the assignment of the metal-binding site in APC2 as a Cys-dependent zinc-binding site. We assume the sub-stoichiometric Zn content of APC2:APC11 results from partial occupancy of APC11 Zn-sites.

Multiple sequence alignments indicate absolute conservation of APC2^ZBM, as defined by the four zinc-coordinating Cys residues within metazoan APC2 sequences (from humans to *C. elegans*) (Fig. 3d), with no Cys conservation in budding yeast, consistent with the absence of a zinc-module in APC2 of the *S. cerevisiae* APC/C cryo-EM structure[30]. The presence of APC2^ZBM correlates well with the conservation of the coactivator N-terminal α-helix in metazoan APC/C coactivator subunits (Figs. 2f, g and 3d). The configuration of the four zinc-coordinating Cys residues of APC2^ZBM is most similar in structure to treble-clef/GATA-like zinc fingers[34].

To determine the role of APC2^ZBM we assessed the consequence of mutating the four zinc-coordinating Cys residues to alanines on APC/C ubiquitination activity. We found that the ubiquitination activity of APC/C, using UBCH10, with the mutant APC2^ZBM (APC/C^APC2ΔZBM) was not impaired relative to wild type APC/C (Supplementary Fig. 4a, c). This result is consistent with the APC2^ZBM and the UBCH10-binding site, at the combined interface of APC2^WHB:APC11^RING, being separated by over 100 Å[18,19] (Fig. 3a).

## Table 1 | Summary of PIXE results

| Wild type APC2:APC11 | | | |
|---|---|---|---|
| **Atom** | **Phosphorus** | **Chlorine** | **Zinc** |
| Atoms/mol | 2.95 ± 0.45 | 1.52 ± 0.26 | 1.90 ± 0.25 |
| LOD (atoms/mol) | 0.64 | 0.34 | 0.19 |
| Mutant APC2^ΔZBM:APC11 | | | |
| Atom | Phosphorus | Chlorine | Zinc |
| Atoms/mol | 4.78 ± 0.48 | 1.37 ± 0.17 | 0.96 ± 0.14 |
| LOD (atoms/mol) | 0.40 | 0.21 | 0.11 |

The number of atoms per molecule of all elements detected with a concentration greater than three times the limit of detection (LOD). The measurements were performed in triplicate from the same protein spot for each sample. The weighted mean values are indicated, with error values as the sample standard deviations. The number of sulphur atoms per molecule was used for internal normalization to obtain the Zn stoichiometry.

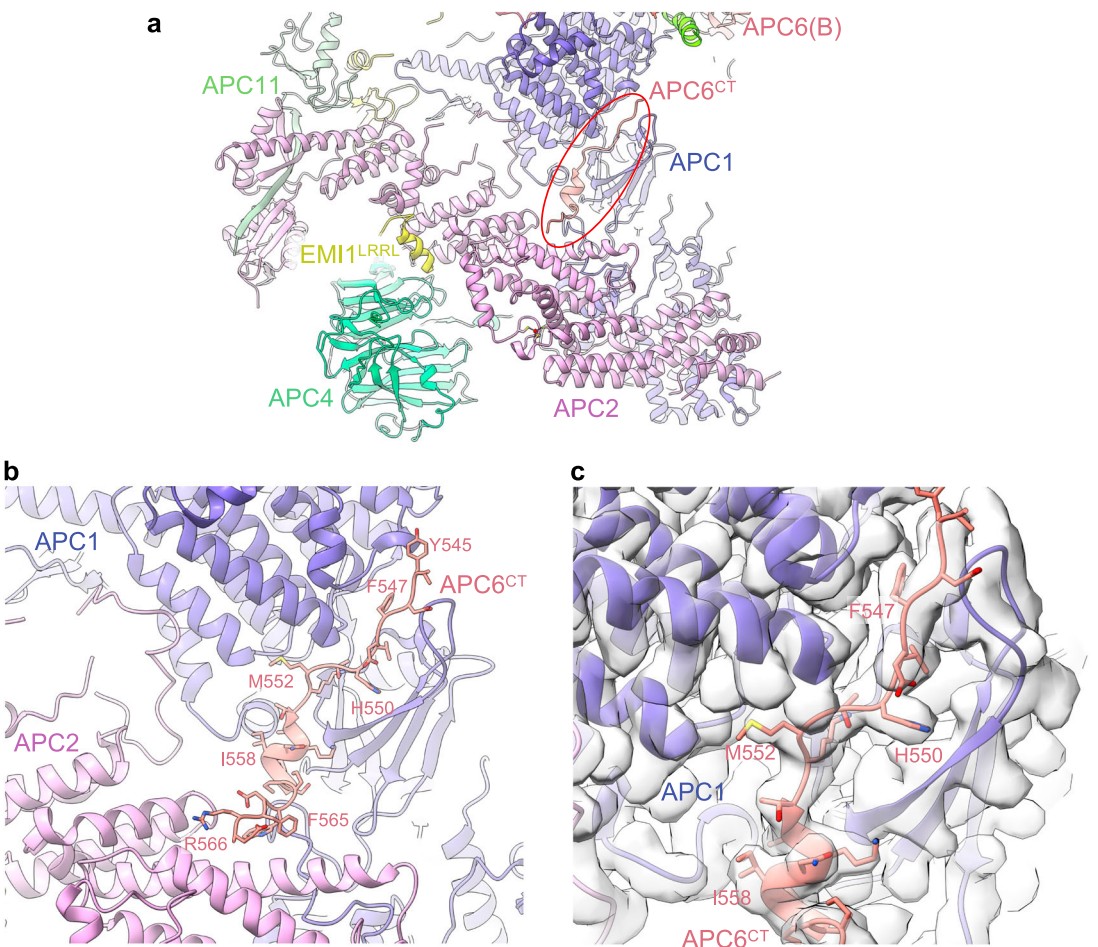

**Fig. 4 | APC6$^{CT}$ - APC1:APC2 inter-connector. a** Overview figure showing APC6$^{CT}$ inter-connector linking the APC1:APC2 interface. **b** Details of the APC6$^{CT}$ APC1-APC2 inter-connector. **c** Cryo-EM density map of the APC6$^{CT}$ APC1:APC2 inter-connector.

We next assessed the consequence of mutating APC2$^{ZBM}$ on APC/C SUMOylation. APC2$^{ZBM}$ is positioned facing towards the APC4 WD40 domain (APC4$^{WD40}$), proximal to the C-terminus of APC4 that includes the two sites of APC/C SUMOylation, Lys772 and Lys798[35] that are disordered in the APC/C cryo-EM maps (Fig. 3a). We considered the possibility that APC2$^{ZBM}$ may contribute to a binding interface for UBC9, the E2 responsible for APC/C SUMOylation[35]. We assessed the consequence of mutating APC2$^{ZBM}$ on the efficacy of APC4 SUMOylation. Compared to wild type APC/C, APC/C with a mutated APC2$^{ZBM}$ (APC/C$^{APC2ΔZBM}$) showed no reduction in APC4 SUMOylation (Supplementary Fig. 4b, c). From this result, we conclude that the APC2$^{ZBM}$ does not contribute to UBC9 recognition by the APC/C. Furthermore, APC2$^{ZBM}$ is remote (>40 Å) from the SUMO- and APC2$^{WHB}$-binding site at the APC2:APC4 interface[35,36] (Fig. 3a), suggesting that APC2$^{ZBM}$ likely does not directly mediate SUMO and APC2$^{WHB}$ binding to the APC/C. Whether the ZBM in APC2 plays a role in APC/C activity under physiological conditions will require future studies.

**Extension of APC6 C-terminus (APC6$^{CT}$)**

In our previous structure (PDB 4UI9), we built an extended region of a chain that transverses the APC1:APC2 subunit interface. The absence of well-resolved side-chain density prevented subunit assignment, although the proximity of one end of this segment to the C-terminal TPR helix of APC6(B) suggested it may correspond to the low-complexity C-terminal region of APC6. In the higher resolution map, clearly defined side-chain density allows assignment to the C-terminal region of APC6(B) (APC6$^{CT}$) (Figs. 1d and 4a–c). Thus, this demonstrates one of the key and reoccurring functions of low complexity/

intrinsically disordered regions present in numerous APC/C subunits, that is, to potentially stabilize inter-subunit interactions through a staple-like mechanism. The equivalent region (residues 544 to 567) is disordered in the second APC6 subunit of the APC6 homodimer APC6(A). We assessed the role of APC6$^{CT}$ in stabilizing APC/C assembly. Truncating APC6 to remove APC6$^{CT}$ had no obvious effect on APC/C assembly. As judged by size exclusion chromatography and SDS PAGE gels (Supplementary Fig. 5a, b), the subunit stochiometry of the purified wild type APC/C and mutant APC/C$^{APCΔ6CT}$ were indistinguishable.

In our studies thus far[19,23], we highlighted the role of the IDRs of the small APC/C subunits APC12, APC13, APC15 and APC16 in forming extended structures that bridge inter-subunit interfaces. APC6$^{CT}$ together with an internal loop of β-blade1 of APC1$^{WD40}$ (residues 35-69) that interacts with APC8(B), exemplify the role of IDRs of larger globular proteins in stabilizing inter-subunit interactions (Fig. 5).

**EMI1 contains a KEN box that interacts with the CDH1 KEN-binding site**

EMI1 inhibits the APC/C through three mechanisms: (i) blockade of the D-box co-receptor through a pseudosubstrate D-box sequence in EMI1 (residues 322 to 330; EMI1$^{D-Box}$), (ii) steric hindrance of UBCH10 binding to the RING domain of APC11, mediated by EMI1$^{ZBR}$, and (iii) through its C-terminal LRRL motif (EMI1$^{LRRL}$), identical to the C-terminus of UBE2S[37,38], EMI1 competes with UBE2S for the LRRL-binding site on APC2, required by UBE2S to bind APC/C[19,38–40] (Figs. 1c and 6a). In our higher resolution maps we noticed cryo-EM density adjacent to the KEN-box binding site of CDH1$^{WD40}$ (Fig. 6b). This density matched closely with the crystal structure of the KEN box of *S. cerevisiae* ACM1

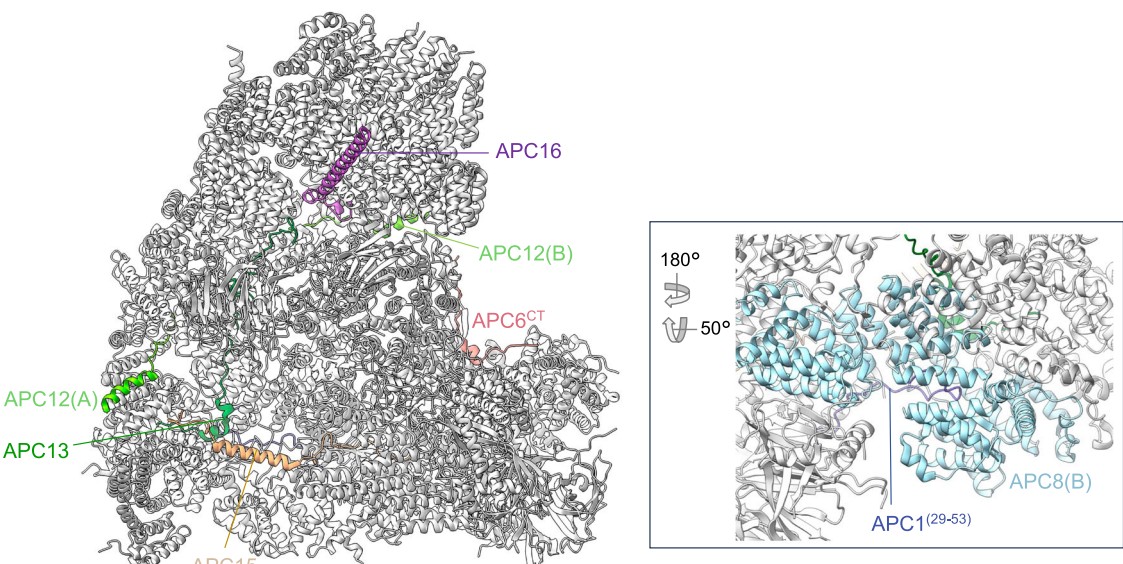

**Fig. 5 | IDRs function to inter-connect APC/C subunits.** Shown coloured are: APC6[CT] (residues 544-567), Apc12, Apc13, Apc15, Apc16. Inset: APC1 (residues 29-53) interacting within the TPR groove of APC8.

bound to *S. cerevisiae* CDH1[WD40] (PDB: 4BH6)[7]. Aided by an AlphaFold prediction (Supplementary Fig. 6a-d), we fitted residues 117 to 122 of EMI1 to this density (Fig. 6a, b). Residues of the 'NKEN' sequence of EMI1 (residues 117-120) conform to a consensus KEN motif sequence[7], and interact with CDH1[WD40]. Previous studies however, showed that the N-terminal region of EMI1 (residues 1-244) is unable in itself to bind APC/C[CDH1], in contrast to the C-terminal segment of residues 299-447[40]. We propose that the EMI1[KEN] motif, by binding to CDH1[WD40], both enhances the affinity of EMI1 for APC/C[CDH1] concomitantly with occlusion of the KEN-box receptor on CDH1, thereby blocking KEN-box dependent substrates from binding the APC/C. Finally, our higher resolution maps also resulted in an improved model of the C-terminal LRRL motif of EMI1 at the APC2:APC4 interface (Fig. 6a).

EMI1 is both a substrate and inhibitor of the APC/C[26]. To assess whether the putative EMI1[KEN] motif (residues 118-120) contributed to EMI1's capacity to bind APC/C[CDH1] as a substrate, we investigated how disrupting EMI1[KEN] affected the degree of ubiquitination of a segment of EMI1 comprising both EMI1[KEN] and EMI1[D-box] motifs (EMI1 residues 105-340: EMI1[105-340]). EMI1[105-340-WT] was effectively ubiquitinated by APC/C[CDH1], whereas disrupting EMI1[KEN] (EMI1[105-340-ΔKEN]) decreased its degree of ubiquitination (Fig. 6c and Supplementary Fig. 6e), consistent with our model that EMI1[KEN] interacts with the CDH1 KEN-box binding site.

### Apo-APC/C

Cryo-EM 3D classification revealed two apo-APC/C classes, representing 28% of the total selected APC/C particles (Supplementary Fig. 2 and Supplementary Table 1). Apo-APC/C was reconstructed to a resolution of 3.2 Å and rebuilt based on the revised 2.9 Å APC/C[CDH1:EMI1] structure (Fig. 7). As expected, as for APC/C[CDH1:EMI1], we observe the zinc-binding module in APC2. Major remodelling involved regions of APC/C that undergo conformational change on conversion from apo-APC/C to the ternary state, promoted by CDH1 binding[23]. These regions comprise the catalytic platform of APC2 and adjacent subunits APC1, APC4, APC5 and APC8(B). The conformational differences between APC/C[CDH1:EMI1] and apo-APC/C involve the downwards movement of the APC2:APC11 module bringing the catalytic module (APC2[CTD]:APC11) to a position close to APC4:APC5 that blocks binding of the ubiquitin-charged UBCH10 to APC2[CTD]:APC11[19]. The conformational change to the active state is induced by the

N-terminal domain of CDH1 (CDH1[NTD]) interacting with the proteosome-cyclosome (PC) domain of APC1 (APC1[PC]). In the ternary complex, CDH1[NTD] binds to a site on APC1[PC] that competes for APC8(B). This disruption of APC1[PC] and APC8(B) interactions induces a downwards displacement of APC8(B), resulting in an upwards movement of the APC2[CTD]:APC11 module (Supplementary Movie 1).

With the higher resolution apo-APC/C map, we re-examined the structure of the APC1[AI], also aided by an AlphaFold prediction of a model of APC8(B) with residues 201-400 of APC1 (Supplementary Fig. 7a-d). The AlphaFold run generated a model, scored with high confidence, predicting that residues 345 to 357 of APC1 dock as an α-helix to the C-box binding site of APC8(B) (APC8(B)[C box]). Leu350 and Arg352 of APC1 are well defined in density (Fig. 8a-c), with the Arg and Leu mimicking the conserved Arg and Ile of the C-box motif (DR[47][F/Y] I[49]PxR) (Fig. 8d). Ser345, Ser351 and Ser355 of APC1, sites of mitotic phosphorylation[41–43], and in vitro phosphorylation by CDK2-cyclin A3-CKS2[9,27], are located within this highly-conserved AI segment (Supplementary Fig. 7e). Phosphorylation of these serines would not be compatible with APC1[AI] binding to APC8(B)[C box], suggesting a mechanism by which APC1 phosphorylation relieves APC8(B)[C-box] inhibition by APC1[AI], thereby allowing CDC20 binding (Fig. 8b inset). Ser355 is within cluster-5 of serine residues of human APC1 (S355, S362, S372, S377) that when mutated to glutamate, promote APC/C[CDC20] activation[31].

To test the model that phosphorylation of Ser345, Ser351 and Ser355 activates APC/C[Cdc20], we substituted glutamates for Ser345, Ser351 and Ser355 of APC1 (APC/C[APC1-3E]) and tested its E3 ligase activity with CDC20 using both phosphorylated and unphosphorylated APC/C and APC/C[APC1-3E]. As we had observed previously[27], unphosphorylated wild type APC/C had very low E3 ligase activity with CDC20 as the coactivator, but was activated by CDK2-cyclin A3-CKS2-mediated phosphorylation of APC/C (Fig. 8e and Supplementary Fig. 8a, b), whereas CDH1 activated both unphosphorylated and phosphorylated APC/C equally well (Fig. 8f and Supplementary Fig. 8a, b). Consistent with the model that Ser345, Ser351 and Ser355 phosphorylation stimulates APC/C[CDC20], APC/C[APC1-3E] was constitutively active, and phosphorylation did not further enhance its activity (Fig. 8e). We note that prior studies had indicated considerable redundancy in the capacity of widely spaced APC1 phospho-sites (within residues 313-380) to stimulate APC/C[CDC20] (refs. 27,31).

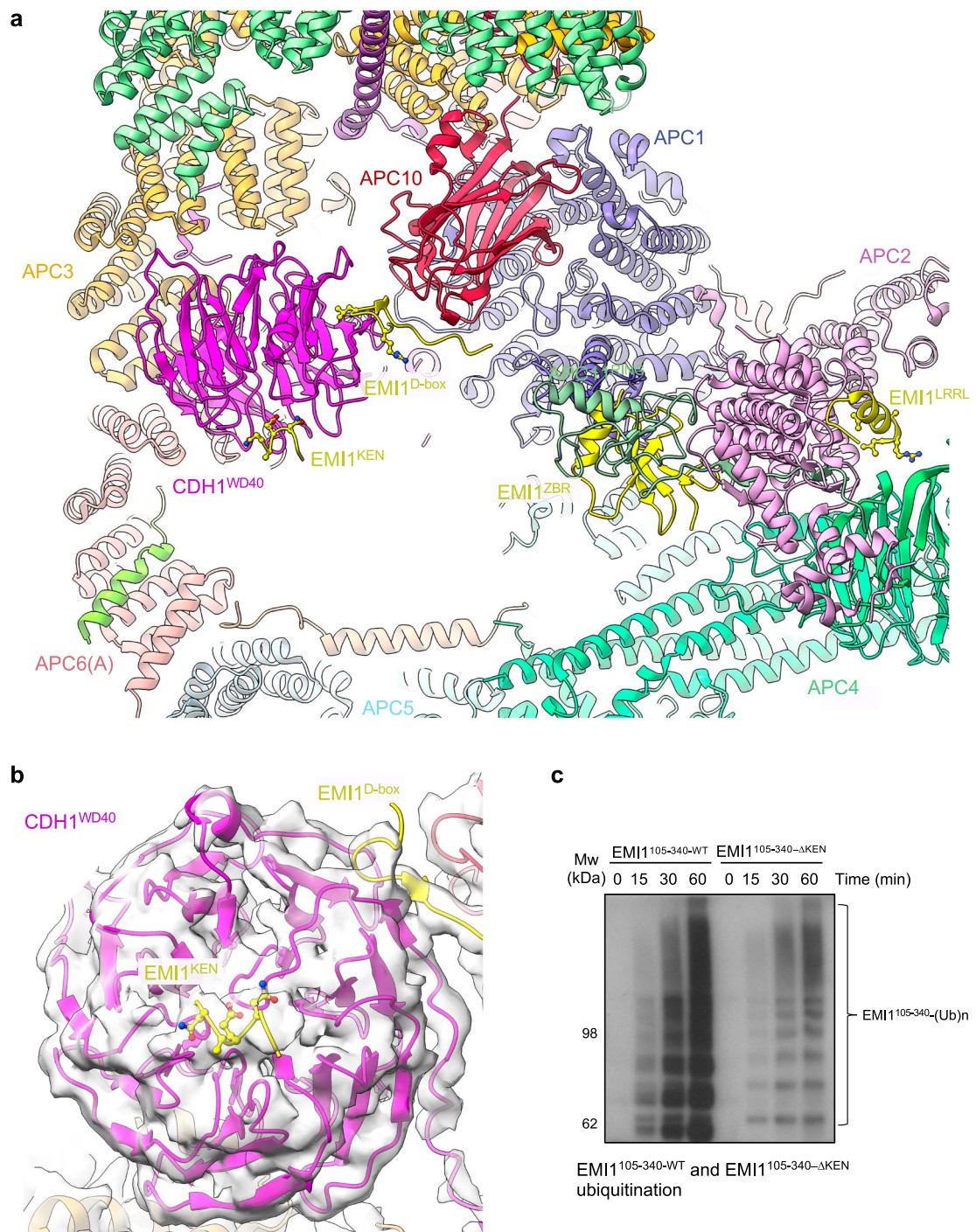

**Fig. 6 | EMI1^{KEN} binding to CDH1. a** EMI1 binds to four sites on the APC/C^{CDH1:EMI1} complex, thereby inhibiting APC/C^{CDH1} ubiquitination activity: (i) EMI1^{KEN} at the KEN-box binding site of CDH1^{WD40}, (ii) EMI1^{D box} at the D-box receptor, (iii) EMI1^{ZBR} to APC11^{RING}, and (iv) EMI1^{LRRL} to the APC2:APC4 interface (PDB: 9GAW). **b** Cryo-EM density map of EMI1^{KEN} binding to CDH1^{WD40} (EMD-19711). Shown are residues NKEN of the KEN motif. The Lys side chain is truncated. **c** Comparison of the ubiquitination of EMI1^{105-340-WT} and EMI1^{105-340-ΔKEN} catalysed by APC/C^{CDH1}. Relative quantities of EMI1^{105-340-WT} and EMI1^{105-340-ΔKEN} are shown in Supplementary Fig. 6e. The experiment show was performed in triplicate. Source data are provided as a Source Data file.

## Discussion

Our earlier 3.6 Å resolution map of the APC/C^{CDH1:EMI1} complex was determined in 2014[19], during the early period of the cryo-EM resolution revolution[44]. The cryo-EM data were collected on a Polara microscope fitted with the first generation of direct electron detectors (FEI Falcon 2). The higher resolution structures reported here benefit from advances in both cryo-EM hardware (Titan Krios

electron microscopes with a more coherent and parallel electron beam), and direct electron detectors with improved detector quantum efficiency (DQE) and data collection rates, coupled to energy filters. These hardware developments have been combined with improvements in data processing analysis in RELION[45] and cryoSPARC[46], allowing the segmentation of flexible large complexes into rigid body domains, combined with signal subtraction methods,

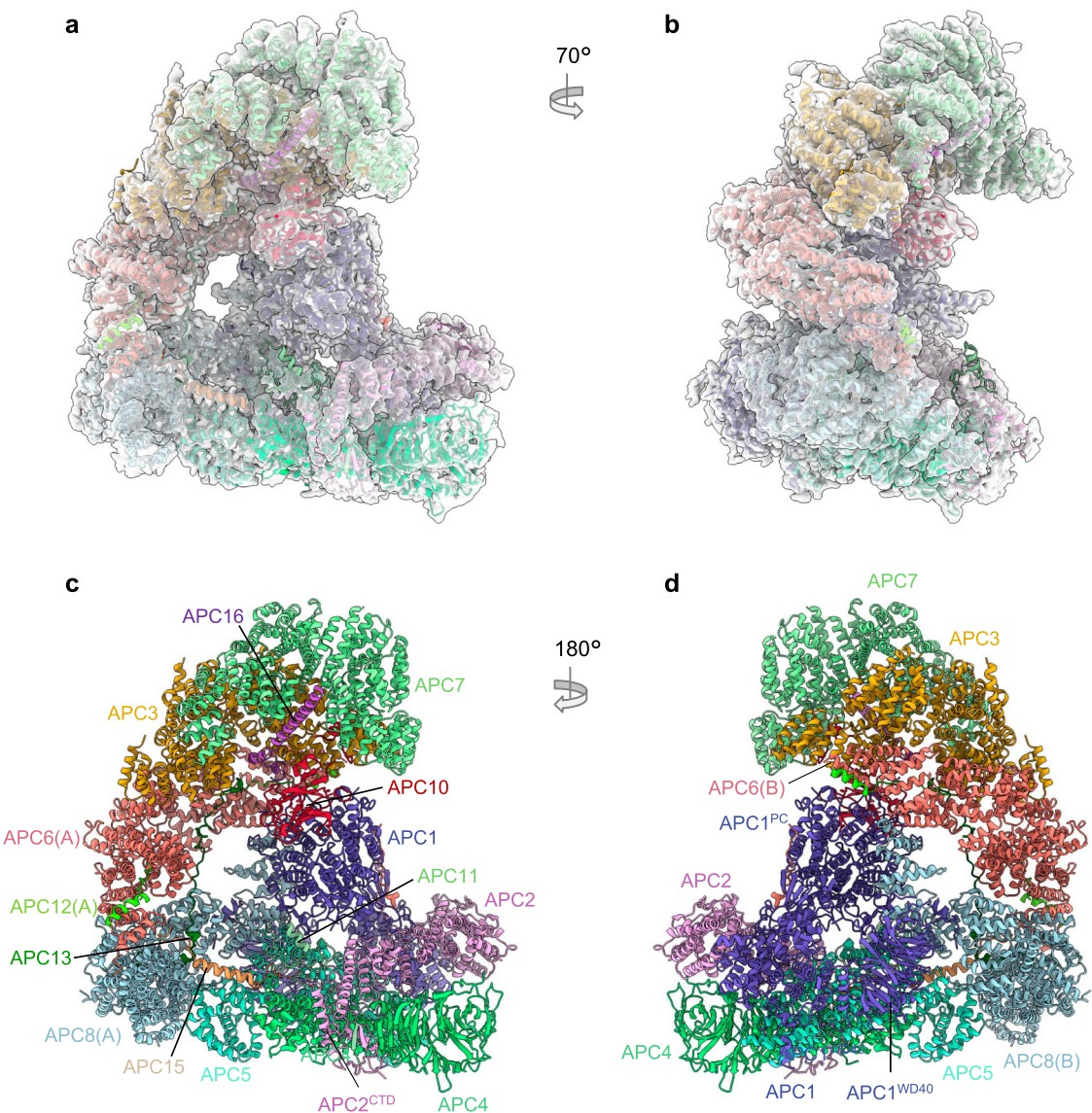

**Fig. 7 | Overall structure of unphosphorylated apo-APC/C. a, b** Two views of apo-APC/C fitted into the 3.2 Å cryo-EM map. **c, d** Two views of apo-APC/C shown as ribbon representations.

for independent alignment, 3D classification and refinement of discrete structural units. Last, the huge advances in the accuracy and reliability of protein structure prediction achieved through AlphaFold[28,29] and RoseTTAFold[47], allow the interpretation of difficult-to-build regions of cryo-EM maps. Cryo-EM grid sample variability, for example more optimal ice thickness on different regions of the grid, may also contribute to the improved data and resultant maps. Altogether these developments have resulted in insights into APC/C structure and function including (i) definition of the N-terminal α-helix of CDH1, (ii) definition at the residue level of previously unbuilt low complexity regions (formed from IDRs), for example the APC6 C-terminus (Supplementary Table 3), (iii) definition of the Zn-binding module of APC2, and (iv) discovery of the pseudo-substrate KEN motif of EMI1 that blocks the KEN-box binding site of CDH1, contributing to EMI1's capacity to be ubiquitinated by APC/C[CDH1], and presumably also EMI1-mediated inhibition of APC/C[CDH1]. This work highlights the enhanced biological insights possible from improving the resolution and refinement of cryo-EM reconstructions originally determined at ~3.5 to 4 Å resolution, aided by AlphaFold predictions of difficult-to-build regions of the cryo-EM density map.

## Methods

### Cloning, expression and purification of recombinant human APC/C, EMI1 and CDH1 and CDC20 mutants

The APC/C[CDH1:EMI1] ternary complex was prepared as described[19,48,49]. Briefly, APC/C[CDH1:EMI1] was co-expressed from two viruses in High 5 insect cells (ThermoFisherScientific, catalogue number B85502). Cells were harvested when the cell viability decreased to 80%. The complex was purified by a combination of Strep-Tactin (Qiagen), anion exchange chromatography Mono Q and Superose 6 size-exclusion chromatography (Cytiva).

To generate the APC/C[APC1-3E] (APC1 subunit with S345E, S351E and S355E mutants), APC/C[APC6ΔCT] (APC6 subunit with residues 544-567 deleted) and APC/C[APC2ΔZBM] (APC2 with ZBM mutated), complexes, the respective APC1, APC6 and APC2 genes were mutated by means of the USER cloning method[50]. The genes for recombinant human APC/C subunits were cloned and assembled into a modified MultiBac system as described[48,49]. This method is based on modified MultiBac pFBDM and pUCDM vectors that allows USER (Uracil-Specific Excision Reagent) ligation-independent cloning[50], generating multi-gene-containing baculovirus transfer vectors for insect cell co-expression. APC/C[APC1-3E], APC/C[APC6ΔCT] and

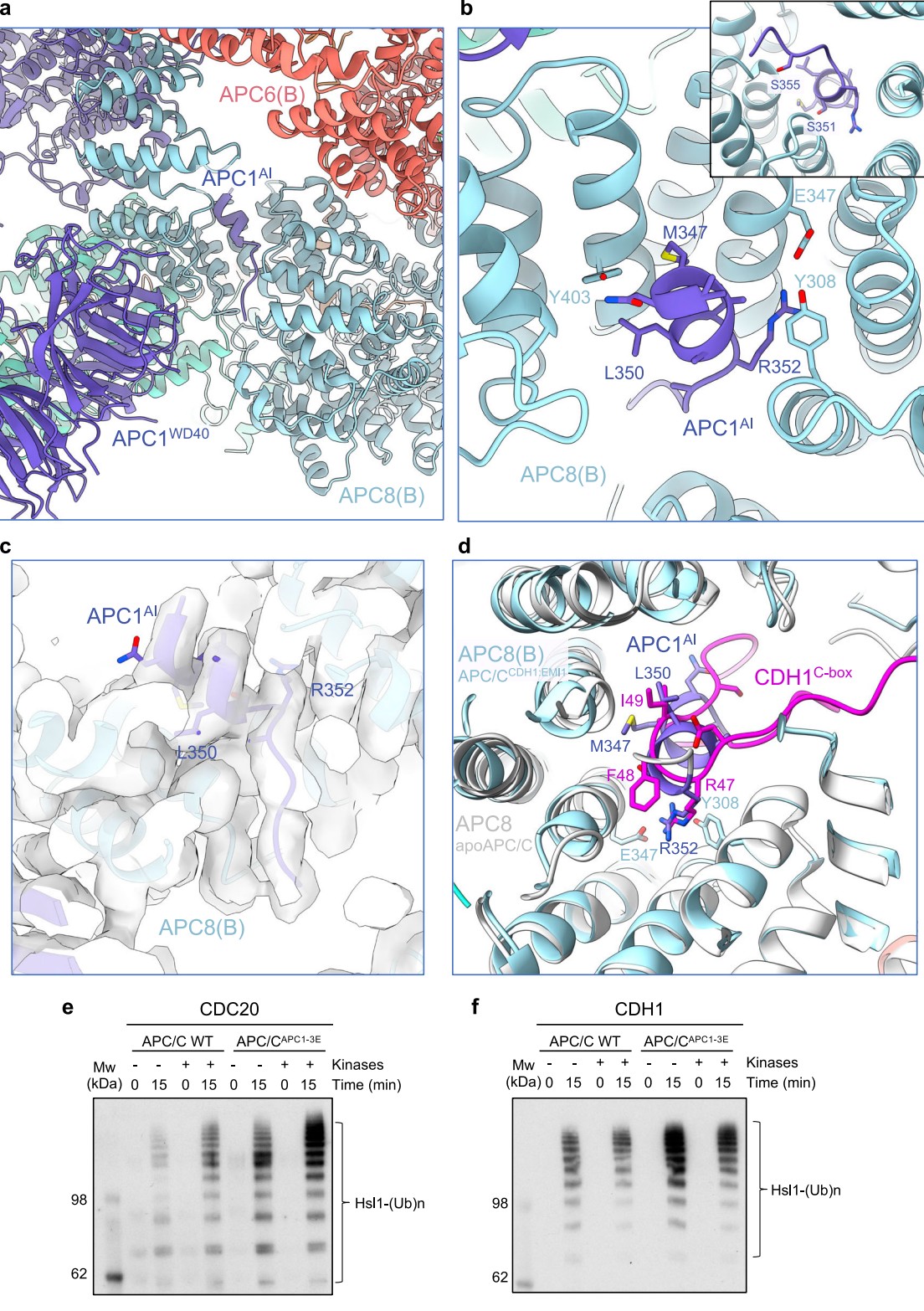

**Fig. 8 | APC1^AI of unphosphorylated apo-APC/C. a** Overall figure showing APC1^AI at the C-box binding site of APC8. **b** Details of APC1^AI interactions at the C-box binding site of APC8. **c** Cryo-EM density map of APC1^AI. **d** Superimposition of the unphosphorylated apo-APC/C (white) onto the APC/C^CDH1:EMI1 complex showing how the APC1^AI (slate blue) and CDH1^C box (magenta) overlap. Glutamates were substituted for Ser345, Ser351 and Ser355 of the AI segment of APC1 (APC/C^APC1-3E) and the activity of unphosphorylated and phosphorylated wild type APC/C and APC/C^APC1-3E

stimulated by CDC20 (**e**) and CDH1 (**f**) were compared. The extent of APC/C phosphorylation was assessed by the more slowly migrating phosphorylated APC3 band on SDS PAGE gels (Supplementary Fig. 8a). The relative quantities of CDC20 and CDH1 used in the ubiquitination assay are shown in (Supplementary Fig. 8b). The experiment show was performed in triplicate. Source data are provided as a Source Data file.

APC/C$^{APC2\Delta ZBM}$ complexes were expressed and purified as for wild type APC/C.

To perform APC/C phosphorylation[27], concentrated APC/C after Resource Q (Cytiva) was treated with CDK2–cyclin A3–CKS2 and PLK1 kinases in a molar ratio of 1:1.5 (APC/C: kinases) in a reaction buffer of 40 mM HEPES pH 8.0, 10 mM MgCl$_2$ and 0.6 mM DTT with 5 mM ATP and 50 mM NaF. The reaction mixture was incubated at 30 °C for 30 min before the final purification step on a Superose 6 3.2/300 column (GE Healthcare) in the APC/C gel-filtration buffer (20 mM HEPES pH 8.0, 150 mM NaCl, 0.5 mM TCEP).

EMI1$^{105-340-WT}$ and EMI1$^{105-340-\Delta KEN}$ (KEN residues 118-120 replaced with Ala) were cloned with a double StrepII-SUMO tag at their N-termini into the pET28 vector (Merck) by means of the USER method[50], and expressed in *E. coli* B834$^{rare2}$ cells. The proteins were purified by loading the lysate onto three tandemly linked Strep-Tactin columns (Qiagen). After thoroughly washing, the protein was eluted with 2.5 mM desthiobiotin (Sigma) in a buffer of 50 mM Tris-HCl (pH 8.0), 200 mM NaCl, 1 mM DTT and 1 mM EDTA. The EMI1 was concentrated and further purified using size exclusion chromatography on a Hiload$^{TM}$ Superdex 200 size exclusion column (Cytiva).

CDH1$^{CDC20\alpha1}$ and CDC20$^{CDH1\alpha1}$ are chimeric constructs of CDH1 and CDC20, respectively with CDH1$^{CDC20\alpha1}$ (α1 helix of CDC20 substituted for α1 of CDH1) and CDC20$^{CDH1\alpha1}$ (α1 helix of CDH1 substituted for α1 of CDC20). CDH1$^{CDC20\alpha1}$: (residues 1 to 23 of CDC20 fused to residues 24 to 493 of CDH1), CDC20$^{CDH1\alpha1}$: (residues 1 to 23 of CDH1 fused to residues 20 to 499 of CDC20). CDH1 and CDC20$^{CDH1\alpha1}$ were fused to an N-terminal His$_6$-hemagglutin antigen (HA) tag. CDC20 and CDH1$^{CDC20\alpha1}$ were fused to an N-terminal His$_6$-MBP-TEV (maltose-binding protein-tobacco etch virus) protease cleavage site tag. All CDH1 and CDC20 and their mutations were expressed in High Five insect cells. To purify CDH1 and its mutants, the insect cells were lysed in a buffer of 50 mM Tris-HCl (pH 7.3), 500 mM NaCl, 20 mM imidazole (buffer A), with a protease inhibitor cocktail. After loading, onto an Ni-NTA column, the column was washed followed by a gradient wash to 200 mM imidazole in buffer A and further washing with ten column volumes of 200 mM imidazole in buffer A. The proteins were eluted in buffer B of 300 mM imidazole (pH 7.3), 500 mM NaCl. CDH1 was finally purified using Superdex 200 size exclusion chromatography in 20 mM HEPES (pH 7.0), 200 mM NaCl, 1 mM DTT. Harvested cell pellets for CDC20 and mutants were resuspended in Cdc20 lysis buffer (50 mM HEPES pH 7.8, 500 mM NaCl, 30 mM imidazole, 10% glycerol and 0.5 mM TCEP) supplemented with 0.1 mM PMSF, 5 units per mL benzonase and Complete EDTA free protease inhibitors and loaded onto a HisTrap HP column (Cytiva). CDC20 was eluted with a gradient to 300 mM imidazole. Collected peak fractions were TEV-cleaved overnight in the dialysis bag (cut-off 6−8 kDa) against the dialysis buffer (50 mM HEPES, pH 7.8, 300 mM NaCl, 5% glycerol and 0.5 mM TCEP) at 4 °C. The protein was re-applied onto the HisTrap HP column and the flow-through was collected.

### APC/C ubiquitination assays

APC/C ubiquitination assays with wild type APC/C, phosphorylated APC/C and the the APC/C$^{APC2\Delta ZBM}$ mutant were performed as described[19,23,27]. The EMI1$^{105-340-WT}$ and EMI1$^{105-340-\Delta KEN}$ substrates were at 20 μM, Hsl1$^{667-872}$ was at 20 μM, and coactivators at 50 μM. Briefly, APC/C (60 nM) and substrate were added to a ubiquitination mix (containing all components except APC/C and substrate) to a 10 μL reaction mix (40 mM Tris-HCl (pH 7.5), 10 mM MgCl$_2$, 0.6 mM DTT, 5 mM ATP, 150 nM UBA1 (E1), 300 nM UBCH10 (E2), 50 μM coactivator, 70 μM His$_6$-ubiquitin, 0.25 mg/ml BSA). Reaction mixtures were incubated at room temperature (22 °C) and terminated by addition of SDS/PAGE loading buffer. Reactions were analysed by SDS/PAGE (8–12% gradient gels) followed by western blotting with an antibody against His$_6$−ubiquitin.

### APC/C SUMOylation assays

For the APC/C SUMOylation assays recombinantly expressed and purified wild type APC/C and the APC/C$^{APC2\Delta ZBM}$ mutant were used as substrates in the presence of Uba2-Aos1 (E1 enzymes), Ubc9, and either SUMO-1 or SUMO-2 and the reaction run for four hours. The SUMOylation reactions were analysed by immunoblotting with an anti-APC4 antibody to detect APC4 SUMOylation, as described[35].

### Cloning and expression of wild type and mutant APC2:APC11 complexes

Wild type APC2:APC11 with an N-terminal double StrepII-tag on APC2 was cloned into the baculovirus expression system using the pUSER method[48]. The mutant APC2:APC11 complex (APC2$^{\Delta ZBM}$:APC11) was generated by substituting Ala for Cys221, Cys224, Cys231 and Cys233 of APC2. Wild type APC2:APC11 and mutant APC2$^{\Delta ZBM}$:APC11 complexes were expressed in High 5 insect cells. Cells were harvested when the cell viability decreased to 80%. Harvested cells were lysed and loaded onto three tandemly linked Strep-Tactin columns (Qiagen) and the complexes were eluted with 2.5 mM desthiobiotin (Sigma) in a buffer of 50 mM Tris-HCl (pH 8.0), 200 mM NaCl, 1 mM DTT. The elution was treated with C-terminal tobacco etch virus (TEV) protease at 4 °C overnight to cleave the double StrepII-tag. The complexes were further purified using Resource Q anion-exchange and size exclusion chromatography (Superdex 200 column) (Cytiva) in a buffer of 20 mM HEPES (pH 8.0), 200 mM NaCl, 2 mM DTT.

### Proton-induced X-ray emission (PIXE) experiments

Wild type and mutant APC2:APC11 (APC2$^{\Delta ZBM}$:APC11 (APC2: C221A,C224A,C231A,C233A)) complexes were dialysed for 16 h at 4 °C against a chloride-free buffer of 50 mM Tris-acetate (pH 7.5) and 200 mM sodium acetate (pH adjusted using acetic acid) and concentrated to 1.6 mg/mL (wild type) and 2.5 mg/mL (APC2$^{\Delta ZBM}$:APC11 mutant). The PIXE experiment was performed at the Ion Beam Centre, University of Surrey, using methods described[33,51]. Briefly, a 350 pA 2.5 MeV proton beam of diameter ~2.0 μm was used to induce characteristic X-ray emission from dried protein complex droplets (volume per droplet ~0.1 μL) under vacuum. The X-rays were detected using a silicon drift detector (Rayspec Ltd) with active area 80 mm$^2$ and energy resolution of 130 eV at 5.9 keV. Spatial maps were obtained of all elements heavier than neon present in the sample by scanning the proton beam over the dried samples. Quantitative information, using sulphur as an internal standard, was obtained by collecting 3 or 4 point spectra from each droplet. Comparison of the quantities of sulphur and zinc allowed the determination of the number of zinc ions in the APC2:APC11 complex.

### Cryo-EM grid preparation and data collection

For cryo-EM grid generation, the purified APC/C$^{CDH1:EMI1}$ complex was diluted to ~200 μg/mL and aliquots of 2 μL were applied to Quantifoil R2/2 grids coated with a second layer of home-made thin carbon film (estimated to be ~5 nm thick), treated with a 9:1 argon:oxygen plasma for 20 s before use. The grids were incubated for 30 s at 4 °C and 100% humidity and then blotted for 6 s and plunged into liquid ethane using an FEI Vitrobot III. Data collection was performed on Krios TEMs equipped with Falcon 3 detectors (ThermoFisherScientific). Details in Supplementary Table 1.

### Cryo-EM image processing

Image processing was performed in cryoSPARC v.3.3.1 and v.4.4.0[46] and RELION 3.1[45]. A detailed processing pipeline can be found in Supplementary Fig. 2. In summary, movies in two separate data sets (a total of 8296 movies) were motion-corrected using MotionCor2[52] (implemented in RELION) and dose-weighted micrographs were imported into cryoSPARC to perform the contrast transfer function

(CTF) parameter estimation using patch-based CTF estimation. Particles were first manually picked on a small subset of micrographs. This particle set was used to train a Topaz model for automated particle picking[53], which was then applied to the entire data set. Particles were cleaned through two rounds of 2D classification. A first ab initio reconstruction with four classes was generated and a subsequent heterogenous 3D refinement was performed and subjected to non-uniform refinement combined with CTF refinement. The resolution was further improved by Bayesian polishing in RELION[45]. After the Bayesian polishing step the resulting particles from the two independent data sets were combined and re-imported into cryoSPARC for another round of 3D classification using eight classes. This allowed efficient separation of APC/C particles in the apo state (Classes 1 and 2) from the intact APC/C$^{CDH1:EMI1}$ particles (Classes 7 and 8). Selected particles were used for a final 3D non-uniform refinement and CTF refinement run to generate a consensus 2.9 Å-resolution map (EMD-13931) (Supplementary Fig. 2 and Supplementary Table 1). Local refinements were performed by applying soft masks on the catalytic domain (Mask 1 – EMD-13732)), on the TPR lobe of APC/C (Mask 2 – EMD-13733), and the central region containing the APC8 as well as CDH1 (Mask 3 – EMD-51070), respectively. For refinements with Mask 1 or 3, particle subtraction had been performed prior to local refinement. In addition, and to further improve the density of CDH1 and the KEN box of EMI1, the CDH1-bound APC/C particles were subjected to another round of 3D classification without alignment by applying a mask covering CDH1. A class containing 54,395 particles showed improved CDH1 density and was subsequently selected for another round of non-uniform refinement and local refinement using Mask 3 (EMD-19711). All final maps were sharpened using deepEMhancer[54]. Final resolution estimation was calculated using gold-standard Fourier Shell Correlation (FSC) based on two independent half maps, and applying the 0.143 criterion[55]. Maps of the consensus APC/C$^{CDH1:EMI1}$ (2.9 Å) ternary complex (EMD-13731), 3D-focussed and refinement map based on CDH1 (EMD-19711), together with maps from Masks 1-3 (EMD-13932, EMD-13933 and EMD-51070, respectively) are are indicated in Supplementary Fig. 2 and Supplementary Table 1. A composite map (EMD-51190) was generated by combining EMD-13731 (consensus 2.9 Å map), EMD-19711 (focussed 3D classification and refinement based on CDH1), and the 3D focussed and refined map of the catalytic module based on Mask 1 (EMD-13732) (Supplementary Fig. 2 and Supplementary Table 1).

For Apo-APC/C, particles selected from classes 1 and 2 of the eight-3D classification (as explained above) were used for a final 3D non-uniform refinement and CTF refinement run to generate the apo-APC/C map (EMD-17751). To improve the quality of the map corresponding to the catatlytic module, particle subtraction was performed by applying a soft mask on the catalytic domain (Mask 4) (Supplementary Fig. 2 and Supplementary Table 1) followed by a 3D classification without alignment. Particles with well resolved density of the catalytic module were selected and subjected to a final 3D non-uniform refinement (EMD-17751-additional map). Particle subtraction had been performed prior to 3D classification and final 3D refinement.

## Model building

Model building of the APC/C$^{CDH1:EMI1}$ ternary complex was performed using COOT[56]. The available structure of APC/C$^{CDH1:EMI1}$ (PDB: 4UI9)[19] was first fitted into the cryo-EM maps in Chimera[57] as an initial model. The model was rebuilt manually based on the higher resolution cryo-EM density. Maps generated using Masks 1, 2 and 3 (Supplementary Fig. 2 and Supplementary Table 1) were also used to guide model building. Undefined regions were built ab initio into the densities. Modelling of difficult to interpret regions of the maps were aided using AlphaFold predictions[28,29]. The final model (PDB: 9GAW) was generated and refined using a composite map (EMD-51190) generated by

combining EMD-13731 (overall 2.9 Å map), EMD-19711 (focussed 3D classification and refinement based on CDH1), and the 3D focussed and refined map of the catalytic module based on Mask 1 (EMD-13732) (Supplementary Figs 1d and 2, and Supplementary Table 1). Model building of the apo-APC/C structure (PDB: 8PKP) was performed identically to the APC/C$^{CDH1:EMI1}$ structure, using the available structure (PDB: 5G05)[27] as an initial model (Supplementary Fig. 1d and Supplementary Table 1). The Alphafold prediction of APC1$^{AI}$ was fitted into the density and refined manually using COOT. Real-space refinement was performed in PHENIX[58] and the refined model was validated using the MolProbity tool[59]. A density fit analysis was performed using COOT[56]. Maps and models were visualized, and figures generated, using Chimera X[60]. Multiple sequence alignments were displayed using JalView2[61].

## AlphaFold2 predictions

AlphaFold2[28] predictions were performed locally. For the APC1$^{AI}$:APC8 model, residues 201 to 400 of APC1 and full-length APC8 were used. The Alphafold prediction was fitted into the density and refined manually using COOT. For the CDH1$^{WD40}$:EMI1$^{KEN box}$ model, full length sequences of CDH1 and EMI1 were used.

PDB and cryo-EM maps have been deposited with RCSB and EMDB, respectively. Accession numbers are listed in Supplementary Table 1. Source data are provided with this paper. Correspondence and requests for materials should be addressed to David Barford.

## Reporting summary

Further information on research design is available in the Nature Portfolio Reporting Summary linked to this article.

## Data availability

PDB coordinates generated in this study have been deposited with RCSB under accession codes 8PKP and 9GAW Cryo-EM maps generated in this study have been deposited with EMDB under accession codes EMD-17751, EMD-51190, EMD-13931), EMD-13932), EMD-13933 ([https://www.ebi.ac.uk/emdb/EMD-13933]), EMD-51070 ([https://www.ebi.ac.uk/emdb/EMD-51070]), EMD-19711 ([https://www.ebi.ac.uk/emdb/EMD-19711]. Previously published protein coordinates used in study: 4UI9 and 5G05. Accession numbers are also listed in Supplementary Table 1. Source data are provided with this paper. Correspondence and requests for materials should be addressed to David Barford. Source data are provided with this paper.

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

## Acknowledgements
We are grateful to the LMB EM Facility for help with the EM data collection, J. Grimmett, T. Darling and I. Clayson for computing, and J. Shi for help with insect cell expression. We thank Stanislau Yatskevich for reagents and A. Andreeva for advice on the APC2$^{ZBM}$ structure. This work was supported by grants of the Swiss National Science Foundation (310030_185235 and MSGI3_211581) (A.B.) and the Swiss Cancer League (KFS-5453-08-2021) (A.B.) and by UKRI/Medical Research Council MC_UP_1201/6 (D.B.) and Cancer Research UK C576/A14109 (D.B.).

## Author contributions
The project has been conceived by D.B, A.B. and L.C. Experiments were performed by A.H., J.Yu, J.Yang., Z.Z., L.C., S.H.M, G.W.G., E.F.G., A.B., D.B. The paper was written by D.B., A.B., A.H., J.Yu, with input from all other authors.

## Competing interests
The authors declare no competing interests.
