## [Transparent Peer Review file · Nature Communications]

Cryo-EM structures of apo-APC/C and APC/C^{CDH1:EMI1} complexes provide insights into APC/C regulation

Corresponding Author: Dr David Barford

Version 0:

Reviewer comments:

Reviewer #1

(Remarks to the Author)

In this manuscript, the authors describe additional structural elements of the anaphase promoting complex (APC). As the APC is a massive multiprotein complex with deep roots in the cell cycle, an accurate and detailed model of its structure is indeed important. Here, the authors describe improved structures for apoAPC and the EMI1, an APC inhibitor, bound state. Specifically, they improve the structures in a few key areas, including the N-terminal helix of CDH1, identity of a ZBM in APC2, KEN box of EMI1, and an IDR of APC6.

While the improved structural models are helpful and will certainly be used to guide other studies, the significance of the new features described is largely unclear. For example, potentially the most interesting finding, in my opinion, is the comparison between the N-terminus of CDH1 and CDC20. However, there is no functional data (biochemical or cellular) to validate their hypothesis that these differences make CDC20 more sensitive to APC1 phosphorylation than CDH1. In summary, for each new feature, either no validation is reported, the authors discuss experiments that are not shown, or they reference unpublished results. Therefore, this study has a lot of potential to be great if these items are addressed.

Other concerns:

The referencing should be improved. For example, lines 60-73 include text describing the relationship between the coactivators, substrates, and E2s. In this section, the authors include only 1 reference, and it is to their own work. The findings of these APC components and how they work together has been the subject of many additional publications.

The manuscript should be reformatted to meet more standard practices.

Reviewer #2

(Remarks to the Author)

Hoefler et al. have presented improved structures of the Anaphase Promoting Complex with a resolution better than 3Å. This stands as one of the most refined structures available to date. Through these enhanced structures, the authors have been able to identify certain densities that were previously unassignable. The manuscript, while offering valuable structural insights into this vital molecular machinery, would benefit from better organization and thorough data presentation. Below are some specific observations:

1. The presented maps and models appear to be of higher quality compared to earlier results from both this group and others. Yet, there are concerns regarding the modeling of certain regions that are not visible in the density, such as FBX5 335-339 and specific parts of CDH1. Additionally, while there is discernable density for APC11, its quality is debatable. In instances where the density is not definitive, especially for side chains, it might be more prudent to omit these details from the model.

2. The authors attribute the enhanced structural quality to advancements in both hardware and software. However, it's arguable whether the previous hardware was indeed a limiting factor for resolution to such a degree, and the new setup doesn't appear to be cutting-edge. Within the resolution range the authors are working, the quality of the sample is typically

the predominant factor, overshadowing the role of the microscope. Thus, the authors may wish to reconsider this claim.

3. On multiple occasions, the authors state that certain tests were conducted, but the corresponding data is missing or not presented. For example, the abstract mentions tests on the zinc-binding capabilities, but the pertinent experimental results are not shown evidently in the main content. Likewise, the sumoylation data seems absent in all figures. For the sake of transparency and validation, all claims should be supported with appropriate data or referenced accordingly. The authors also state in the statistics form that any statistical analysis is not applicable in their case. If they indeed have conducted experiments to test their claims statistics is something that should have been used.

In its current form, the manuscript primarily enumerates newly discerned densities without weaving them into a cohesive narrative. While I haven't come across a manuscript of such nature before, the quality of the structure is undeniably an advancement for the field. It may prove invaluable for future structural endeavors, potentially serving as a reference. The decision on its suitability for publication remains at the discretion of the editor.

Version 1:

Reviewer comments:

Reviewer #1

(Remarks to the Author)

The revised manuscript is improved with additional experiments describing the relative contributions of the new structural features. Therefore, they have largely addressed my concerns. However, the authors need to fix the legend of Supplementary Figure 3 as it does not match the figure shown.

Reviewer #2

(Remarks to the Author)

The manuscript improved with more experiments which unfortunately are not giving many new insights. Thus, the main point remains the higher resolution structure, only. Still, the new structure may be an important resource for future studies of the APC and has . Whether this is enough for publication in Nature communications and whether the unusual list style is acceptable I leave at the editors discretion. However, I see still some issues especially regarding model building that should be addressed.

Major Points:

1. Unfortunately, I don't recall the details of the previous models, so I cannot comment on the extent of improvement. However, there are still several regions lacking discernible density, and the authors are inconsistent in their treatment of those. In some cases, these regions are deleted in the model, while in others, they are modeled. I would highly recommend to only model visible parts. Additionally, there are regions where the density is insufficient for modeling side chains, which should be removed. Here are a few examples:

in pdb 8S4G:

- APC2 in its front part is smeary due to its well-known flexibility, which only allows for rigid body interpretation. Side chains should not be modeled here. The same applies to parts of APC11. Although the density may have improved, it's still not sufficient for definitive modeling.

- APC2 308-316 has hardly any density. In other instances, the authors have removed the model in such cases. Consistency in approach is needed.

- CDC23 (chain U) lacks density for 555-559, but it was previously cut until 524, so please cut 524-559.

- CDH1 shows a frame shift (e.g., 297-308), and side chains are not well-aligned with the density.

- The zinc-binding motif of APC2, which is a major focus, is not well modeled. The cysteines are not well within the density, and the Zn is not nicely coordinated. The authors should pay more attention to these key points.

High-resolution model 7QE7:

- CDH1 is modeled differently in large parts compared to the other model, however the density is insufficient to justify this.

- APC7 521-537 is shifted compared to the density.

I suggest performing a density fit analysis in Coot to verify which amino acids fit the density and which do not, or using Q-scores throughout. Given that the structure is the manuscript's main selling point, it should be as precise as possible.

2. It's unclear how the focused maps were used in model building and why they are not deposited in the EMDB.

3. The thermostability assay raises some questions: The assignment of melting points to the two proteins seems speculative

and could be incorrect. The 43.8 peak is still visible as a shoulder in the delta zbm mutant (which is incorrectly labeled in the figure). Thus the new peak maybe something else. This analysis is delicate, especially for protein complexes. While still a missing Zn will affect stability, the authors fail to highlight the significance this stability.

4. I still disagree that the improved structure is likely due to software and hardware advances. The authors argue that the sample is the same, but it cannot be identical; it would at least be another region on the grid or likely another grid, where ice thickness plays a role. The previously used setup was not limiting within this resolution range, and the software and hardware used are even old from today's stand point. If the authors' claim is correct, using the latest software should further improve the structure. In this sense to make really a leap forward I suggest that the authors follow their claim and use the most modern software to further improve their structure.

Minor Points:

5. The authors should define delta zbm in the main text as the tetra mutant (e.g., in line 209ff), not only in the methods, to avoid confusion.

6. The processing scheme for the apo map is missing image and particle numbers, making it hard to follow.

Version 2:

Reviewer comments:

Reviewer #1

(Remarks to the Author)

The manuscript is improved from its original submission as stated previously. The structure is an improvement over existing models.

Reviewer #2

(Remarks to the Author)

Thank you for your revisions. The map and model presented are satisfactory and could warrant publication. While there may be some remaining issues, it's worth noting that any structure of this complexity may have minor errors and inconsistencies. However, it's important to highlight that the paper offers limited new biological insights beyond the structure itself. I'll leave it to the editor's discretion to determine if this is sufficient for publication in Nature Communications.

Regarding the thermal stability assays, I must respectfully disagree with their interpretation. While these assays are indeed valuable for analyzing small single-domain proteins, their application to complexes and multi-domain proteins is more challenging. The literature shows mixed results: some studies suggest limited utility for protein complexes, while others demonstrate potential usefulness under specific conditions. The latter cases, however, typically require two-state transition curves, which are not present in your data.

The melting of the APC likely involves numerous pathways that are difficult to interpret definitively. It's crucial to recognize that melting curves of complexes are not simply linear combinations of individual component melting curves. Your data shows multiple transitions, and assigning specific meanings to these transitions would be speculative at best. Given these considerations, I would recommend removing this section entirely to maintain the overall strength and reliability of your findings.

Reviewer #1:

Remarks to the Author:

In this manuscript, the authors describe additional structural elements of the anaphase promoting complex (APC). As the APC is a massive multiprotein complex with deep roots in the cell cycle, an accurate and detailed model of its structure is indeed important. Here, the authors describe improved structures for apoAPC and the EMI1, an APC inhibitor, bound state. Specifically, they improve the structures in a few key areas, including the N-terminal helix of CDH1, identity of a ZBM in APC2, KEN box of EMI1, and an IDR of APC6.

While the improved structural models are helpful and will certainly be used to guide other studies, the significance of the new features described is largely unclear. For example, potentially the most interesting finding, in my opinion, is the comparison between the N-terminus of CDH1 and CDC20. However, there is no functional data (biochemical or cellular) to validate their hypothesis that these differences make CDC20 more sensitive to APC1 phosphorylation than CDH1. In summary, for each new feature, either no validation is reported, the authors discuss experiments that are not shown, or they reference unpublished results. Therefore, this study has a lot of potential to be great if these items are addressed.

We thank the reviewer for carefully and critically reading the manuscript, and for suggesting important new experiments and text changes that have improved the manuscript.

We agree with the value of testing and validating the new structural features and hypothesis resulting from the higher resolution determination of the APC/C^{CDH1.EMI1} and apo-APC/C complex structures. Specifically, in the revised manuscript we include the following new studies.

- 1. N-terminal α -helices ($\alpha 1$) of CDH1 and CDC20.** We tested the roles of the N-terminal α -helices ($\alpha 1$) of CDH1 and CDC20 by comparing both unphosphorylated and phosphorylated APC/C ubiquitination activity using a CDH1 chimera, swapping the respective $\alpha 1$ -helices (CDH1 ^{$\alpha 1$} and CDC20 ^{$\alpha 1$}). Whereas the CDH1 mutant with the CDC20 $\alpha 1$ -helix (CDH1^{CDC20 $\alpha 1$}) was stable, the CDC20 mutant with CDH1 ^{$\alpha 1$} was unstable. We therefore compared the activities of unphosphorylated and phosphorylated APC/C activated by CDH1 and CDH1^{CDC20 $\alpha 1$} . The CDH1^{CDC20 $\alpha 1$} coactivator stimulated the same ubiquitination activity for both unphosphorylated and phosphorylated APC/C, similar to CDH1 (**Supplementary Fig. 3a, b**). This indicated that the $\alpha 1$ helix is not a primary factor in determining the higher activity of CDH1 for unphosphorylated APC/C relative to CDC20. We showed previously that differences between CDH1 and CDC20 in both their N-terminal domains and IR tails (excluding the WD40 domain) contribute to the higher affinity of CDH1 for unphosphorylated APC/C¹.
- 2. Zinc-binding module of APC2 (APC2^{ZBM}).**
 - We now include the experimental results discussed in the original manuscript testing the effects of mutating the zinc-binding module of APC2 (APC2^{ZBM}) on APC/C ubiquitination and SUMOylation activities (**Supplementary Fig. 4**).
 - Additionally, we tested the thermal stability of the wild type APC2:APC11 and APC2 ^{Δ ZBM}:APC11 heterodimers, finding that the mutant APC2 ^{Δ ZBM}:APC11

heterodimer is 8°C less thermal stable than the wild type. This indicated that the zinc-binding module of APC2 contributes to its thermal stability above 37°C (Fig. 3e).

3. **APC6^{CT}**. The C-terminus of one subunit of the APC6 homodimer transverses the APC1:APC2 subunit interface (APC6^{CT}). We proposed that this stabilizes APC/C assembly. However, a mutant of APC/C with APC6^{CT} deleted appeared to assemble as well as wild type APC/C as judged by SDS PAGE analysis and size exclusion chromatography (Supplementary Fig. 5).
4. **KEN box of EMI1**. To test the model that the newly discovered KEN box motif on EMI1 binds to CDH1, we compared APC/C^{CDH1}-mediated ubiquitination of a fragment of wild type EMI1 and EMI1^{ΔKEN} (residues 105-340). Disruption of the KEN box substantially reduced levels of EMI1 ubiquitination by APC/C^{CDH1}, consistent with the model that the KEN box of EMI1 interacts with the KEN-box binding site on CDH1 (Fig. 6c and Supplementary Fig. 6e).
5. **Auto-inhibitory segment of APC1 (APC1^{AI})**. The newly refined structure of apo-APC/C indicated that APC1^{AI} spans residues 345 to 357, that includes Ser345, Ser351 and Ser355. Ser355 is within cluster-5 of serine residues of human APC1 (S355, S362, S372, S377) that when mutated to glutamate, promote APC/C^{CDC20} activation, mimicking APC/C phosphorylation². We tested the consequences of substituting glutamates for all three Ser residues (Ser345, Ser351 and Ser355). The mutant complex (APC/C^{APC1-3E}) was constitutively activated by CDC20. This activity was only slightly increased by APC/C^{APC1-3E} phosphorylation (Fig. 8e,f and Supplementary Fig. 8).

The authors discuss experiments that are not shown, or they reference unpublished results. Therefore, this study has a lot of potential to be great if these items are addressed.

In addition to the experiments referred to above, on two occasions we used ‘data not shown’ when referring to the absence of sequence similarity between metazoan and budding yeast at the N-terminus of CDH1, and between the zinc-binding module of APC2, thus indicating that budding yeast lacks these structural features. We have deleted ‘data not shown’. We also used ‘unpublished results’ when referring to our deposited structure of the budding yeast APC/C^{CDH1.HSL1} complex (deposited at RCSB). Although we have not published the manuscript describing this structure, the coordinates and cryo-EM maps are available to the community via RCSB and EMDB data bases (8A3T and EMD-15123, respectively). Thus we should not have cited these cords as ‘unpublished results’ and we have removed this, and all other instances of ‘data not shown’ and ‘unpublished results’.

Other concerns:

The referencing should be improved. For example, lines 60-73 include text describing the relationship between the coactivators, substrates, and E2s. In this section, the authors include only 1 reference, and it is to their own work. The findings of these APC components and how they work together has been the subject of many additional publications.

The manuscript should be reformatted to meet more standard practices.

We apologise for this. The revised manuscript more extensively, and we hope accurately, cites the work of other groups in a more balanced manner.

Reviewer #2:

Remarks to the Author:

Hoefler et al. have presented improved structures of the Anaphase Promoting Complex with a resolution better than 3Å. This stands as one of the most refined structures available to date. Through these enhanced structures, the authors have been able to identify certain densities that were previously unassignable. The manuscript, while offering valuable structural insights into this vital molecular machinery, would benefit from better organization and thorough data presentation. Below are some specific observations:

We thank the reviewer for carefully and critically reading the manuscript, and for suggesting important new experiments and text changes that have improved the manuscript.

1. The presented maps and models appear to be of higher quality compared to earlier results from both this group and others. Yet, there are concerns regarding the modeling of certain regions that are not visible in the density, such as FBX5 335-339 and specific parts of CDH1. Additionally, while there is discernable density for APC11, its quality is debatable. In instances where the density is not definitive, especially for side chains, it might be more prudent to omit these details from the model.

We thank the reviewer for this comment. To further improve the density of CDH1 and the KEN box of EMI1, the CDH1-bound APC/C particles were subjected to another round of 3D classification without alignment by applying a mask covering CDH1. A class containing 54,395 particles showed improved CDH1 density and was subsequently selected for another round of non-uniform refinement and local refinement using Mask 3. The main-chain and many side chains of CDH1 are now resolved. As suggested by the reviewer, we deleted residues 335-339 of EMI1 (FBX5) (and also residues 111-119 of CDH1). Most of the side chains of APC11 are accounted for in density. However, we did replace residues with ambiguous or residual density at the periphery with stubs. We deposited the revised coordinates and map at RCSB (8S4G), and EMDB (EMD-19711), respectively. Supplementary Table 1 is updated, and also made similar revisions to coordinates 7QE7.

2. The authors attribute the enhanced structural quality to advancements in both hardware and software. However, it's arguable whether the previous hardware was indeed a limiting factor for resolution to such a degree, and the new setup doesn't appear to be cutting-edge. Within the resolution range the authors are working, the quality of the sample is typically the predominant factor, overshadowing the role of the microscope. Thus, the authors may wish to reconsider this claim.

This is an interesting suggestion. However, the sample used in this study, and cryo-EM grid preparation, were identical to that used in the Chang *et al.* (2015) paper³. Thus we think that it is likely that the major contribution to the improved maps is a result of advances in hardware and

software.

3. On multiple occasions, the authors state that certain tests were conducted, but the corresponding data is missing or not presented. For example, the abstract mentions tests on the zinc-binding capabilities, but the pertinent experimental results are not shown evidently in the main content. Likewise, the sumoylation data seems absent in all figures. For the sake of transparency and validation, all claims should be supported with appropriate data or referenced accordingly.

In the original manuscript the PIXE data for identifying the metal ions at the APC2 zinc-binding site of APC2 were presented in Supplementary Table 3. To clarify these data, we now include this table in the main text (**Table 1**).

As suggested by the reviewer, we now include the results of mutating the zinc-binding site of APC2 on APC/C ubiquitination and SUMOylation activities (**Supplementary Fig. 4**).

Additionally, we performed new experiments to test hypothesis generated from the structure. Specifically, in the revised manuscript we include the following new studies. These are outlined in detail in response to reviewer 1. In summary these include:

1. Testing the roles of the N-terminal α -helices ($\alpha 1$) of CDH1 and CDC20 on the capacity of CDH1 and CDC20 to activate unphosphorylated APC/C (**Supplementary Fig. 3a, b**). This experiment indicated that the $\alpha 1$ helix is not a primary factor in determining the higher activity of CDH1 for unphosphorylated APC/C relative to CDC20.
2. Zinc-binding module of APC2 ($APC2^{ZBM}$).
 - a. Inclusion of APC/C ubiquitination and SUMOylation activities of the $APC2^{ZBM}$ mutant (**Supplementary Fig. 4**) and an additional experiment testing the effects of mutating $APC2^{ZBM}$ on the thermal stability of APC2 (**Fig. 3e**).
3. We tested whether deleting the C-terminus of APC6 affected APC/C assembly. According to our assay, deletion of the APC6 C-terminus had no detectable effect on APC/C assembly (**Supplementary Fig. 5**).
4. KEN box of EMI1. We showed that mutating the KEN box of EMI1 substantially decreased the efficiency of APC/C^{CDH1}-mediated ubiquitination of an N-terminal fragment of EMI1 (**Fig. 6c and Supplementary Fig. 6e**).
5. Auto-inhibitory segment of APC1 ($APC1^{AI}$). To test the model that phosphorylation of the Ser residues of $APC1^{AI}$ stimulates CDC20-mediated activation of the APC/C, we replaced these Ser residues with Glu. The resultant mutated APC/C was constitutively activated by CDC20. This activity was only slightly increased by APC/C phosphorylation (**Fig. 8e,f and Supplementary Fig. 8**).

The authors also state in the statistics form that any statistical analysis is not applicable in their

case. If they indeed have conducted experiments to test their claims statistics is something that should have been used.

Statistical analyses were applied to the zinc stoichiometry of APC2 (**Table 1**) and the thermal unfolding temperature of wild type APC2:APC11 and the APC2^{DZBM}:APC11 mutant. We indicate the mean and standard deviations and the number of repeats. We have now indicated this information in Reporting Summary form.

In its current form, the manuscript primarily enumerates newly discerned densities without weaving them into a cohesive narrative. While I haven't come across a manuscript of such nature before, the quality of the structure is undeniably an advancement for the field. It may prove invaluable for future structural endeavors, potentially serving as a reference. The decision on its suitability for publication remains at the discretion of the editor.

We hope we have addressed the concerns of the referees.

- 1 Zhang, S. *et al.* Molecular mechanism of APC/C activation by mitotic phosphorylation. *Nature* **533**, 260-264, doi:10.1038/nature17973 (2016).
- 2 Qiao, R. *et al.* Mechanism of APC/CCDC20 activation by mitotic phosphorylation. *Proceedings of the National Academy of Sciences of the United States of America* **113**, E2570-2578, doi:10.1073/pnas.1604929113 (2016).
- 3 Chang, L., Zhang, Z., Yang, J., McLaughlin, S. H. & Barford, D. Atomic structure of the APC/C and its mechanism of protein ubiquitination. *Nature* **522**, 450-454, doi:10.1038/nature14471 (2015).

REVIEWER COMMENTS

Reviewer #1 (Remarks to the Author):

Our response in red.

The revised manuscript is improved with additional experiments describing the relative contributions of the new structural features. Therefore, they have largely addressed my concerns. However, the authors need to fix the legend of Supplementary Figure 3 as it does not match the figure shown.

We thank the reviewer for this positive feedback and for spotting this error in the Supplementary Figure 3 legend. Now corrected.

Reviewer #2 (Remarks to the Author):

Our response in red.

The manuscript improved with more experiments which unfortunately are not giving many new insights. Thus, the main point remains the higher resolution structure, only. Still, the new structure may be an important resource for future studies of the APC and has . Whether this is enough for publication in Nature communications and whether the unusual list style is acceptable I leave at the editors discretion. However, I see still some issues especially regarding model building that should be addressed.

Major Points:

1. Unfortunately, I don't recall the details of the previous models, so I cannot comment on the extent of improvement. However, there are still several regions lacking discernible density, and the authors are inconsistent in their treatment of those. In some cases, these regions are deleted in the model, while in others, they are modeled. I would highly recommend to only model visible parts. Additionally, there are regions where the density is insufficient for modeling side chains, which should be removed.

Here are a few examples:

in pdb 8S4G:

- APC2 in its front part is smeary due to its well-known flexibility, which only allows for rigid body interpretation. Side chains should not be modeled here. The same applies to parts of APC11. Although the density may have improved, it's still not sufficient for definitive modeling.
- APC2 308-316 has hardly any density. In other instances, the authors have removed the model in such cases. Consistency in approach is needed.
- CDC23 (chain U) lacks density for 555-559, but it was previously cut until 524, so please cut 524-559.

In the previous review round, the reviewer raised concerns regarding density for CDH1 and EMI1 (FBX5, residues 335-339) in map EMD-13931 (PDB:7QE7). As we explained in our response then, we calculated a new map using 3D focussed refinement based on a mask covering CDH1 (EMD-19711, PDB:8S4G). Therefore the comments of the reviewer regarding PDB:8S4G on APC2, CDC23 (chain U), and the zinc-binding site in APC2 seem inappropriate because in this map, to improve density for the flexible CDH1, other regions of the map will be less clear. We detail our response to specific comments on PDB:8S4G below.

Similarly, the reviewer's comment: '(High-resolution model 7QE7:- CDH1 is modeled differently in large parts compared to the other model [presumably PDB:8S4G], however the density is insufficient to justify this.)' was addressed in our revised manuscript with new maps and coordinates (EMD-19711, PDB:8S4G). We are unclear why the reviewer raised this again.

To avoid possible confusion over the purpose of the two sets of maps and coordinates (EMD-13931/PDB:7QE7 and EMD-19711/PDB:8S4G) and to further address questions regarding the fit of coordinates, we have performed the following: We generated a new composite cryo-EM map, and also deleted regions of the APC/C coordinates that are not supported by well-resolved cryo-EM density in this composite map. Specifically we generated a composite map (EMD-51190) that combined three maps: EMD-13931 (the original consensus 2.9 Å map), EMD-19711 (the map generated from the 3D class with higher CDH1 occupancy in response to the first round of reviews), and the map from Mask 1 (focussed 3D classification and refinement of the catalytic module subunit APC2 – EMD-13932) (**Supplementary Figs 1d and 2a, and Supplementary Table 1**). In this new composite map, both the EM densities for CDH1 and the catalytic module, that are less well defined in EMD-13931, are better resolved.

We also truncated side chains where density does not extend beyond C-beta, and also some main-chain regions in APC2 and others as defined below.

The new composite map and associated coordinates have accession numbers: are EMD-51190 and PDB:9GAW. These coordinates also apply to EMD-19711 (map with improved density for CDH1, **Supplementary Fig. 2a and Supplementary Table 1**). Thus PDB:9GAW supersedes PDB:7QE7 and PDB:8S4G which will be removed from RCSB. We believe this strategy eliminates any inconsistencies in modelling mentioned by the reviewer.

Our specific response to other reviewer 2 comments.

in pdb 8S4G:

- APC2 in its front part is smeary due to its well-known flexibility, which only allows for rigid body interpretation. Side chains should not be modeled here. The same applies to parts of APC11. Although the density may have improved, it's still not sufficient for definitive modeling.

- APC2 308-316 has hardly any density. In other instances, the authors have removed the model in such cases. Consistency in approach is needed.

The issues with the APC2 density are addressed in the new map and coordinates (EMD-51190 and PDB:9GAW), but as stated above PDB:8S4G/EMD-19711 were generated to optimise CDH1 density.

- CDC23 (chain U) lacks density for 555-559, but it was previously cut until 524, so please cut 524-559.

This region has been deleted in the new map and coordinates (EMD-51190 and PDB:9GAW).

- CDH1 shows a frame shift (e.g., 297-308), and side chains are not well-aligned with the density.

The side chains of CDH1 297-308 are not well defined and have been truncated to stubs in the new map and coordinates (PDB:9GAW).

- The zinc-binding motif of APC2, which is a major focus, is not well modeled. The cysteines are not well within the density, and the Zn is not nicely coordinated. The authors should pay more attention to these key points.

We have improved the Zn ion position and adjusted cysteine side chain conformations (EMD-51190 and PDB:9GAW).

High-resolution model 7QE7:

- CDH1 is modeled differently in large parts compared to the other model, however the density is insufficient to justify this.

As stated above, this point was addressed in our first revision resulting in the new maps EMD19711 and coordinates PDB:8S4G in response to the first round of reviews based on EMD13931 (PDB: 7QE7).

- APC7 521-537 is shifted compared to the density.

We have deleted - APC7 521-537 in (EMD-51190 and PDB:9GAW).

I suggest performing a density fit analysis in Coot to verify which amino acids fit the density and which do not, or using Q-scores throughout. Given that the structure is the manuscript's main selling point, it should be as precise as possible.

We performed a density fit analysis in Coot as suggested.

2. It's unclear how the focused maps were used in model building and why they are not deposited in the EMDB.

Maps generated using Masks 1, 2 and 3 (**Supplementary Fig. 2a and Supplementary Table 1**) were used to aid model building.

Depositing maps. This is a good point. We have deposited maps based on Masks -1, -2 and -3, as EMD-13932, EMD-19333 and EMD-51070, respectively (**Supplementary Figs 1d and 2a, and Supplementary Table 1**).

3. The thermostability assay raises some questions: The assignment of melting points to the two proteins seems speculative and could be incorrect. The 43.8 peak is still visible as a shoulder in the delta zbm mutant (which is incorrectly labeled in the figure **A minor error caused by the file conversion on submission to the *Nature Communication* web site. It is correct in the original submitted file**).

We observe a T_m of 52 °C and 43.8 °C for wild type APC2:APC11, and 52 °C and 35.7 °C for the Zn-binding site mutant. The '43.8 peak' the reviewer comments on is not present in multiple repeats of mutant APC2:APC11 (Figure 1 below). It is reasonable to propose that the 43.8 °C peak present in wild type and not mutant APC2:APC11, and the 35.7 °C peak present in the mutant and not wild type APC2:APC11 correspond to the T_m s for wild type and mutant APC2, respectively. The 52 °C peak remains constant for wild type and mutant APC2:APC11, thus we assign this to APC11, which is the same in both samples.

This analysis is delicate, especially for protein complexes. **We don't understand this comment. Thermostability assays are accepted assays in the community.**

While still a missing Zn will affect stability, the authors fail to highlight the significance this stability.

We have no comment to this.

4. I still disagree that the improved structure is likely due to software and hardware advances. The authors argue that the sample is the same, but it cannot be identical; it would at least be another region on the grid or likely another grid, where ice thickness plays a role. The previously used setup was not limiting within this resolution range, and the software and hardware used are even old from today's stand point. If the authors' claim is correct, using the latest software should further improve the structure. In this sense to make really a leap forward I suggest that the authors follow their claim and use the most modern software to further improve their structure.

The 3.6 Å structure of the APC/C was determined before the introduction of focussed 3D refinement and multibody refinement (Chang et al., 2015). Without doubt these new processing methods have improved our most recent maps. However, we do accept that the quality of the cryo-EM grids can vary, and the cryo-EM grids used for the new data set may have been superior, i.e. more optimal ice thickness etc. on a different region of the grid. We have added this point to the discussion. There have been no new major advances in data processing methods since we completed the processing reported in this manuscript, except for more automated data-processing pipelines. We therefore have not carried out a lengthy reprocessing of all the data.

Minor Points:

5. The authors should define delta zbm in the main text as the tetra mutant (e.g., in line 209ff), not only in the methods, to avoid confusion.

This has been done. We thank the reviewer for the suggestion.

6. The processing scheme for the apo map is missing image and particle numbers, making it hard to follow.

Now added to Supplementary Fig. 2b.

Wild type APC2:APC11

Mutant APC2 Δ ZBM:APC11

Figure 1. Thermal stability of the wild type APC2:APC11 dimer and mutant APC2 Δ ZBM:APC11 dimer. First derivative of the 350/330 ratio. Maxima in the plot indicate T_m where 50% of the protein has unfolded. The experiment was performed 12 times for the wild type APC2:APC11 dimer (blue curves) and nine times for the mutant APC2 Δ ZBM:APC11 dimer (red curves) from the same sample.

REVIEWER COMMENTS

Reviewers' comments in black.

Our response in red.

Reviewer #1 (Remarks to the Author):

The manuscript is improved from its original submission as stated previously. The structure is an improvement over existing models.

We are grateful to the reviewer for reviewing this paper.

Reviewer #2 (Remarks to the Author):

Thank you for your revisions. The map and model presented are satisfactory and could warrant publication. While there may be some remaining issues, it's worth noting that any structure of this complexity may have minor errors and inconsistencies.

However, it's important to highlight that the paper offers limited new biological insights beyond the structure itself. I'll leave it to the editor's discretion to determine if this is sufficient for publication in Nature Communications.

Regarding the thermal stability assays, I must respectfully disagree with their interpretation. While these assays are indeed valuable for analyzing small single-domain proteins, their application to complexes and multi-domain proteins is more challenging. The literature shows mixed results: some studies suggest limited utility for protein complexes, while others demonstrate potential usefulness under specific conditions. The latter cases, however, typically require two-state transition curves, which are not present in your data.

The melting of the APC likely involves numerous pathways that are difficult to interpret definitively. It's crucial to recognize that melting curves of complexes are not simply linear combinations of individual component melting curves. Your data shows multiple transitions, and assigning specific meanings to these transitions would be speculative at best. Given these considerations, I would recommend removing this section entirely to maintain the overall strength and reliability of your findings.

We have removed the thermal stability results. However, we determined T_m for the APC2:APC11 heterodimer, not the complete APC/C complex.